# Distillation ≈ Early Stopping? Harvesting Dark Knowledge Utilizing Anisotropic Information Retrieval For Overparameterized Neural Network

## Abstract

Distillation is a method to transfer knowledge from one model to another and often achieves higher accuracy with the same capacity. In this paper, we aim to provide a theoretical understanding on what mainly helps with the distillation. Our answer is "early stopping". Assuming that the teacher network is overparameterized, we argue that the teacher network is essentially harvesting dark knowledge from the data via early stopping. This can be justified by a new concept, Anisotropic Information Retrieval (AIR), which means that the neural network tends to fit the informative information first and the non-informative information (including noise) later. Motivated by the recent development on theoretically analyzing overparameterized neural networks, we can characterize AIR by the eigenspace of the Neural Tangent Kernel(NTK). AIR facilities a new understanding of distillation. With that, we further utilize distillation to refine noisy labels. We propose a self-distillation algorithm to sequentially distill knowledge from the network in the previous training epoch to avoid memorizing the wrong labels. We also demonstrate, both theoretically and empirically, that self-distillation can benefit from more than just early stopping. Theoretically, we prove convergence of the proposed algorithm to the ground truth labels for randomly initialized overparameterized neural networks in terms of $\ell_2$ distance, while the previous result was on convergence in 0-1 loss. The theoretical result ensures the learned neural network enjoy a margin on the training data which leads to better generalization. Empirically, we achieve better testing accuracy and entirely avoid early stopping which makes the algorithm more user-friendly.

## 1 Introduction

Deep learning achieves state-of-the-art results in many tasks in computer vision and natural language processing LeCun et al. (2015). Among these tasks, image classification is considered as one of the fundamental tasks since classification networks are commonly used as base networks for other problems. In order to achieve higher accuracy using a network with similar complexity as the base network, distillation has been proposed, which aims to utilize the prediction of one (*teacher*) network to guide the training of another (*student*) network. In Hinton et al. (2015), the authors suggested to generate a soft target by a heavy-duty teacher network to guide the training of a light-weighted student network. More interestingly, Furlanello et al. (2018); Bagherinezhad et al. (2018) proposed to train a student network parameterized identically as the teacher network. Surprisingly, the student network significantly outperforms the teacher network. Later, it was suggested by Zagoruyko & Komodakis (2016a); Huang & Wang (2017); Czarnecki et al. (2017) to transfer knowledge of representations, such as attention maps and gradients of the classifier, to help with the training of the student network. In this work, we focus on the distillation utilizing the network outputs Hinton et al. (2015); Furlanello et al. (2018); Yang et al. (2018a); Bagherinezhad et al. (2018); Yang et al. (2018b).

To explain the effectiveness of distillation, Hinton et al. (2015) suggested that instead of the hard labels (*i.e* one-hot vectors), the soft labels generated by the pre-trained teacher network provide extra information, which is called the "Dark Knowledge". The "Dark knowledge" is the knowledge

encoded by the relative probabilities of the incorrect outputs. In Hinton et al. (2015); Furlanello et al. (2018); Yang et al. (2018a), the authors pointed out that secondary information, *i.e* the semantic similarity between different classes, is part of the "Dark Knowledge", and Bagherinezhad et al. (2018) observed that the "Dark Knowledge" can help to refine noisy labels. In this paper, we would like to answer the following question: **can we theoretically explain how neural networks learn the Dark Knowledge?** Answering this question will help us to understand the regularization effect of distillation.

In this work, we assume that the teacher network is overparameterized, which means that it can memorize all the labels via gradient descent training Du et al. (2018b;a); Oymak & Soltanolkotabi (2018); Allen-Zhu et al. (2018). In this case, if we train the overparameterized teacher network until convergence, the network's output coincides exactly with the ground truth hard labels. This is because the logits corresponding to the incorrect classes are all zero, and hence no "Dark knowledge" can be extracted. Thus, we claim that **the core factor that enables an overparameterized network to learn "Dark knowledge" is early stopping**.

What's more, Arpit et al. (2017); Rahaman et al. (2018); Xu et al. (2019) observed that "Dark knowledge" represents the discrepancy of convergence speed of different types of information during the training of the neural network. Neural network tends to fit informative information, such as simple pattern, faster than non-informative and unwanted information such as noise. Similar phenomenon was observed in the inverse scale space theory for image restoration Scherzer & Groetsch (2001); Burger et al. (2006); Xu & Osher (2007); Shi & Osher (2008). In our paper, we call this effect Anisotropic Information Retrieval (AIR).

With the aforementioned interpretation of distillation, We further utilize AIR to refine noisy labels by introducing a new self-distillation algorithm. To extract anisotropic information, we sequentially extract knowledge from the output of the network in the previous epoch to supervise the training in the next epoch. By dynamically adjusting the strength of the supervision, we can theoretically prove that the proposed self-distillation algorithm can recover the correct labels, and empirically the algorithm achieves the state-of-the-art results on Fashion MNIST and CIFAR10. The benefit brought by our theoretical study is twofold. Firstly, the existing approach using large networks Li et al. (2019); Zhang & Sabuncu (2018) often requires a validation set to early terminate the network training. However, our analysis shows that our algorithm can sustain long training without overfitting the noise which makes the proposed algorithm more user-friendly. Secondly, our analysis is based on an $\ell_2$-loss of the clean labels which enables the algorithm to generate a trained network with a bigger margin and hence generalize better.

## 1.1 CONTRIBUTIONS

We summarize our contributions as follows

- This paper aims to understand distillation theoretically (*i.e.* understand the regularization effect of distillation). Distillation works due to the soft targets generated by the teacher network. Based on the observation that the overparameterized network can exactly fit the one-hot labels which contain no dark knowledge, we theoretically justify that early stopping is essential for an overparameterized teacher network to extract dark knowledge from the hard labels. This provides a new understanding of the regularization effect of distillation.

- This is the first attempt to theoretically understand the role of distillation in noisy label refinery using overparameterized neural networks. Inspired by Li et al. (2019), we utilize distillation to propose a self-distillation algorithm to train a neural network under label corruption. The algorithm is theoretically guaranteed to recover the unknown correct labels in terms of the $\ell_2$-loss rather than the previous 0-1-loss. This enables the algorithm to generate a trained network whose output has a bigger margin and hence generalizes better. Furthermore, our algorithm does not need a validation set to early stop during training, which makes it more hyperparameter friendly. The theoretical understanding of the overparameterized networks encourages us to use large models which empirically produce better results.

## 2 Distillation ≈ Early Stopping?

### 2.1 No Early Stopping, No Dark Knowledge

As mentioned in the introduction, an overparameterized teacher network is able to extract dark knowledge from the on-hot hard labels because of early stopping. In this section we present an experiment to verify this effect of early stopping, where we use a big model as a teacher to teach a smaller model as the student.

In this experiment, we train a WRN-28 Zagoruyko & Komodakis (2016b) on CIFAR100 Krizhevsky & Hinton (2009) as the teacher model, and a 5-layers CNN as the student. The experimental details are in the supplementary material. The teacher model was trained by 40, 80, 120, 160 and 200 epochs respectively. The results of knowledge distillation by the student model is shown in Fig. 1.

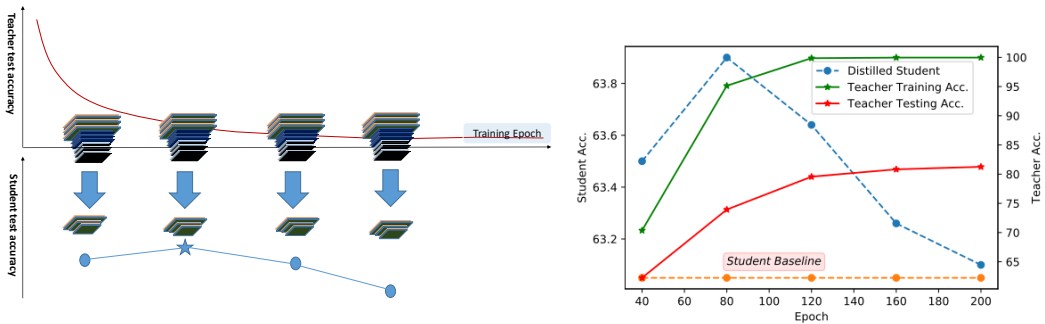

Figure 1: A good teacher may not be able to produce a good student. To maximize the effectiveness of distillation, you should early stop your epoch at a proper time.

As we can see, the teacher model does not suffer from overfitting during the training, while it does not always educate a good student model either. As suggested by Yang et al. (2018a), a more tolerant teacher educates better students. The teacher model trained with 80 epochs produces the best result which indicates that early stopping of the bigger model can extract more informative information for distillation.

### 2.2 Anisotropic Information Retrieval

In this paper, we introduce a new concept called **Anisotropic Information Retrieval (AIR)**, which means to exploit the discrepancy of the convergence speed of different types of information during the training of an overparameterized neural network. An important observation of AIR is that informative information tends to converge faster than non-informative and unwanted information such as noise.

Selective bias of an iterative algorithm to approximate a function has long been discovered in different areas. For example, Xu (1992) observed that iterative linear equation solver fits the low frequency component first, and they proposed a multigrid algorithm to exploit this property. In image processing, Scherzer & Groetsch (2001); Burger et al. (2006); Xu & Osher (2007); Shi & Osher (2008) proposed inverse scale space methods that recover image features earlier in the iteration and noise comes back later. In kernel learning, early stopping of gradient descent is equivalent to the ridge regression Smale & Zhou (2007); Yao et al. (2007); Vito et al. (2005), which means that bias from the eigenspace corresponding to the larger eigenvalues is reduced quicker by the gradient descent.

Under the neural network setting, Rahaman et al. (2018); Xu et al. (2019) observed that neural networks find low frequency patterns more easily. Cicek et al. (2018) discovered that noisy labels can slow down the training. Arpit et al. (2017); Rolnick et al. (2017) studied the memorization effects of the deep networks, and revealed that, during training, neural networks first memorize the data with clean labels and later with the wrong labels. In the next subsection, we will characterize AIR of overparametrized neural networks using the Neural Tangent Kernel Jacot et al. (2018).

## 2.3 AIR and Neural Tangent Kernel

In Jacot et al. (2018), the authors introduced the *Neural Tangent Kernel* to characterize the trajectory of gradient descent algorithm learning infinitely wide neural networks. Denote $f(\theta, x) \in \mathbb{R}$ as the output of neural network with $\theta \in \mathbb{R}^n$ being the trainable parameter and $x \in \mathbb{R}^p$ the input. Consider training the neural network using the $\ell_2$-loss on the dataset $\{(x_i, y_i)\}_{i=1}^n \in \mathbb{R}^p \times \mathbb{R}$:

$$\ell(\theta) = \frac{1}{2} \sum_{i=1}^n (f(\theta, x_i) - y_i)^2.$$

We use the gradient descent $\theta_{t+1} = \theta_t - \eta \nabla l(\theta_t)$ to train the neural network. Let $u_t = (f(\theta_t, x_i))_{i \in [n]} \in \mathbb{R}^n$ be the network outputs on all the data $\{x_i\}$ at iteration $t$ and $y = (y_i)_{i \in [n]}$. It was shown by Du et al. (2018b); Oymak & Soltanolkotabi (2018); Li et al. (2019) that the evolution of the error $u_t - y$ can be formulated in a quasi-linear form,

$$(u_t - y) = (I - \eta \hat{H}_t) \cdot (u_t - y),$$

where

$$\hat{H}_t = \left( \left\langle \int_0^1 \frac{\partial f(\theta_t + \alpha(\theta_{t+1} - \theta_t), x_i)}{\partial \theta} \mathrm{d}\alpha, \frac{\partial f(\theta_t, x_j)}{\partial \theta} \right\rangle \right)_{i,j}.$$

It is known that $\hat{H}_t - H_t = o(1)$ with respect to $d$, where $H_t = \left( \left\langle \frac{\partial f(\theta_t, x_i)}{\partial \theta}, \frac{\partial f(\theta_t, x_j)}{\partial \theta} \right\rangle \right)_{i,j}$ is an $n \times n$ positive semi-definite Gram matrix and $d$ is the width of the neural network Oymak & Soltanolkotabi (2018); Li et al. (2019). This means that $\hat{H}_t = H_t$ when the neural network is infinitely wide. It was further shown by Oymak & Soltanolkotabi (2018); Li et al. (2019); Du et al. (2018a) that when the neural network is infinitely wide, the Gram matrix is static, i.e. $H_t = H^*$. The static Gram matrix $H^*$ is referred to as the *Neural Tangent Kernel*(NTK)Jacot et al. (2018).

Note that $H^*$ is a symmetric positive semi-definite matrix. Assume that $\lambda_1 > \cdots > \lambda_n \geq 0$ are its $n$ eigenvalues and $e_1, e_2, \cdots, e_n$ are the corresponding eigenvectors. The eigenvectors are orthogonal $< e_i, e_j >= 0$ and $H^* = \sum_{i=1}^n \lambda_i e_i e_i^T$. Consider the evolution of the projection of the loss function in different eigenspaces

$$\langle (u_t - y), e_i \rangle = \langle (I - \eta H^*)(u_t - y), e_i \rangle = \langle (u_t - y), (I - \eta H^*) e_i \rangle = (1 - \eta \lambda_i) \langle (u_t - y), e_i \rangle.$$

We can see that the component lies in the eigenspace with a larger eigenvalue converges faster. Therefore, AIR describes the phenomenon that the gradient descent algorithm searches for information components corresponding to different eigenspaces at different rates. Rahaman et al. (2018); Arpit et al. (2017); Arora et al. (2019); Zhang & Sabuncu (2018) has shown that that one possible reason of neural network's good generalization property is that neural network fits useful information faster. Thus in our paper, we regard informative information as the eigenspaces associated with the largest few eigenvalues of NTK.

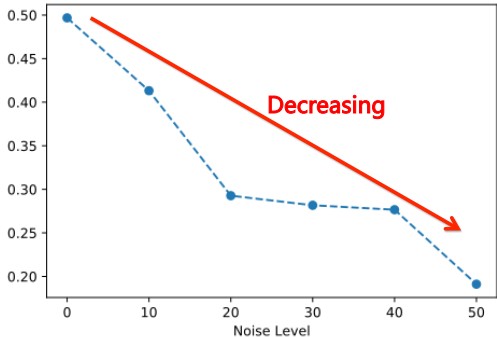

Figure 2: Components of label noise in the largest five eigensapces of NTK are decreasing.

We denote the projection of supervision signal to the eigenspace with a larger eigenvalue as useful information. In Figure 2, We calculate the ratio of the172norm of the label vector provided by the

self-distillation algorithm lies in the top-5 eigenspace as a representative of informative information. We can see that the informative information decreases when the noise level increases. This motivates us to further explore how "Dark Knowledge" helps with label refinery.

# 3 Noisy Label Refinery

Supervised learning requires high quality labels. However, due to noisy crowd-sourcing platforms Khetan et al. (2017) and data augmentation pipeline Bagherinezhad et al. (2018), it is hard to acquire entirely clean labels for training. On the other hand, successive deep models often have huge capacities with millions or even billions of parameters. Such huge capacity enables the network to memorize all the labels, right or wrong Zhang et al. (2016), which makes learning deep neural networks with noisy labels a challenging task. Arpit et al. (2017); Rolnick et al. (2017); Han et al. (2018) pointed out that neural networks often fit the clean labels before the noisy ones during training. Recently, Li et al. (2019) theoretically showed that early stopping can clean up label noise with overparametrized neural networks. This, together with our understanding of distillation with AIR, inspired us to use distillation for noisy label refinery. We shall introduce a new self-distillation algorithm with theoretically guaranteed recovery of the clean labels under suitable assumptions.

## 3.1 Related Works

Training deep models on datasets with label corruption is an important and challenging problem that has attracted much attention lately. In Ma et al. (2018), the authors proposed to regularize the Local Intrinsic Dimensionality of deep representations to detect label noise. Tanaka et al. (2018) proposed a joint optimization framework to simultaneously optimize the network parameters and output labels. Zhang & Sabuncu (2018) introduced a loss function generalizing the cross entropy for robust learning. The approach that is most relevant to ours is called learning with a mentor. For example, Jiang et al. (2017) used a mentor network to learn the curriculum for the student network. Han et al. (2018); Yu et al. (2019) introduced two networks that can teach each other to reject wrong labels. Hu et al. (2019) designed a new regularizer to restrict every weight vector to be close to its initialization for all iterations thus enforcing the same regularization effect as early stopping.

In the literature of distillation, Bagherinezhad et al. (2018) first utilized distillation to refine noisy labels of ImageNet, while Yang et al. (2018b) introduced a distillation method to complete teacher-student training in one generation. The latter is most related to our proposed self-distillation algorithm. However, the difference is that their model aims to ensemble diverse models in one training trajectory but ours aims to utilize AIR to refine noisy labels during the training.

## 3.2 The Self-distillation Algorithm

It is known that the label noise lies in the eigenspaces associated to small eigenvalues Arora et al. (2019); Li et al. (2019). Thus, Li et al. (2019) used early stopping to remove label noise. However, early stopping is hard to tune and sometimes leads to unsatisfactory results. In this section, we proposed a self-distillation algorithm with an excellent empirical performance and a theoretical guarantee to recover the correct labels under certain conditions but without the requirement of early stopping.

From the perspective of AIR, we observe that the knowledge learned in early epochs is informative information (*i.e.* the eigenspaces associated with the largest few eigenvalues of NTK) and can be used to refine the training for later epochs. In other words, the algorithm distills knowledge sequentially to guide the training of the model in later epochs by the knowledge distilled by the model from earlier epochs. The informative information learned during early epochs is, in some sense, "low frequency information", which is the core factor to enable the model to generalize well. A nice property of the self-distillation algorithm is that it generates the final model in one generation (*i.e.* single-round training), which has almost no additional computational cost compared to normal training. The proposed self-distillation algorithm is given by Algorithm 1.

Here, the function $h(\cdot)$ in the algorithm is the label function. It can either be a hard label function such as *hardmax*, or a soft label function such as *softmax* with a certain temperature. The choice of $h(\cdot)$ and interpolation coefficient $\alpha_t$ depends on the usage of the self-distillation algorithm. If

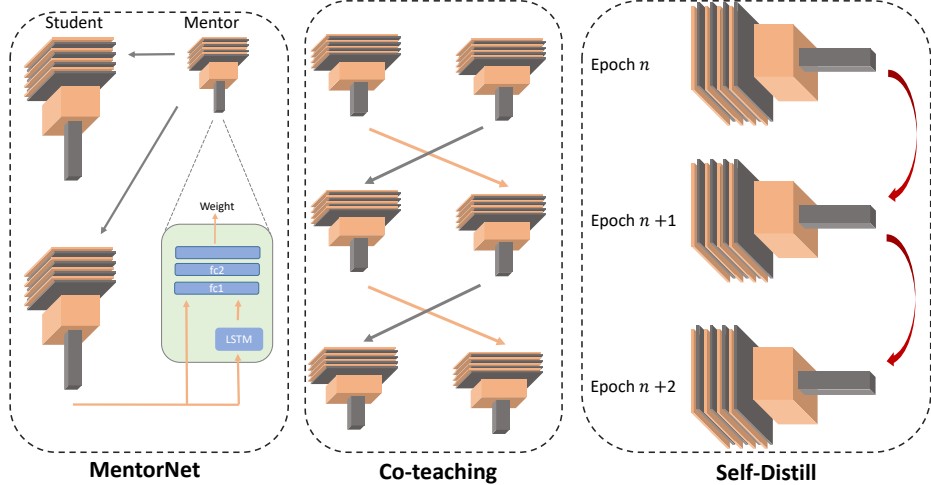

Figure 3: Comparison of error flow among MentorNet Jiang et al. (2017), Co-teaching Han et al. (2018) and our algorithm. Our algorithm does not require another teacher network and hence does not impose any additional computation burden.

---

**Algorithm 1** Self-Distillation

---

Randomly initialize the network. $t = 0$
**repeat**
    Fetch data $(x_1, y_1), \cdots, (x_n, y_n)$ from training set.
    Set the label $\hat{y}_{i,t} = \alpha_t y_i + (1 - \alpha_t) h(\mathcal{N}(x_i, \omega_t))$
    Detach $\hat{y}_i$ from the computational graph
    Update $\omega_{t+1} = \omega_t - \eta \sum_{i=1}^{n} \nabla_\omega l(\mathcal{N}(x_i, \omega_t), \hat{y}_{t,i})$.
    $t = t + 1$
**until** training converged =0

---

we want to clean up label noise, we normally choose $h(\cdot)$ to be *hardmax* or *softmax* with a low temperature. The weight $\alpha_t$ is chosen to be adaptively decreasing corresponding to the increase of our confidence on the learned model at current epoch. The introduction of $h(\cdot)$ helps to boost AIR and the information gained from the previous epoch.

### 3.3 THEORETICAL FOUNDATION OF SELF-DISTILLATION

In this section, we provide a theoretical justification of the performance of the self-distillation algorithm with overparameterized neural networks. Here, we only consider binary classification task with label $\in \{-1, +1\}$.

**Definition 1.** *(Noisy Clusterable Dataset DescriptionsLi et al. (2019))*

- *We consider a dataset with $n$ data: $\{(x_i, y_i, \tilde{y}_i)\}_{i=1}^{n} \in \mathbb{R}^d \times \{-1, +1\} \times \{-1, +1\}$. $(x_i, y_i)$ are the input data and its associated label seen by the model while $\tilde{y}_i$ is the unobserved ground truth label. The pair $(x_i, y_i)$ with $y_i = \tilde{y}_i$ is called a clean data, otherwise it is called a corrupted data.*

- *We assume that $\{x_i\}_{i \in [n]}$ contains points with unit Euclidean norm and has $K$ clusters. The data is randomly i.i.d sampled from a distribution $P$. Let $n_l$ be the number of points in the lth cluster. In our analysis, we ssume that number of data in each cluster is balanced in the sense that $n_l \geq c_{low} \frac{n}{K}$ for constant $c_{low} > 0$ and all of the data is bounded, i.e. $\|x_i\|_2^2 \leq R, \forall i \in [n]$.*

- *For each of the $K$ clusters, we assume that all the input data lie within the Euclidean ball $\mathcal{B}(c_l, \epsilon)$, where $c_l$ is the center with unit Euclidean norm and $\epsilon > 0$ is the radius.*

- *Assume that the data in the same cluster has the same ground truth label $\tilde{y}$. For the lth cluster, we denote $\rho_l$ the proportion of the data with wrong labels. Let $\rho = \max\{\rho_i : i \in [K]\}$ and assume that $\rho < \frac{1}{2}$.*

- *A dataset satisfying the above assumptions is called an $(\epsilon, \rho)$ dataset.*

The above definition of dataset follows that of the previous work Li et al. (2019); Li & Liang (2018). It is reasonable to assume that $\rho < \frac{1}{2}$ in order to ensure the correct labels dominate each cluster. In this work, we consider two-layers neural networks. For input data $x \in \mathbb{R}^d$, the output of the neural network $f$ is:

$$f(W, x) = v^T \phi(Wx),$$

where $W \in \mathbb{R}^{k \times d}$ is the weight matrix and $\phi$ is the activation function applied to $Wx$ entry-wise. We suppose $k$ is even and fix the output layer by assigning half of the entries $\frac{1}{\sqrt{k}}$ and the others $-\frac{1}{\sqrt{k}}$. Given a data matrix $X = [x_1, x_2, \ldots, x_n]^T$, we simply denote the output vector on the data matrix as

$$f(W, X) = [v^T \phi(WX^T)]^T = (f(W, x_1), f(W, x_2), \ldots, f(W, x_n))^T \in \mathbb{R}.$$

Following the previous work on training overparameterized neural network Du et al. (2018b), we consider the MSE loss

$$\mathcal{L}(W, X) = \frac{1}{2}\|f(W, X) - y\|_2^2$$

.

**Definition 2.** *For a data matrix $D \in^{m \times d}$, we denote $\lambda(D)$ the small eigenvalue of the neural network covariance matrix*

$$\Sigma(D) = (DD^T) \odot \mathbb{E}_{g \sim \mathcal{N}(0, I_d)}[\phi'(Dg)\phi'(Dg)^T].$$

The above definition reveals the matching score of the model and data. We denote $C = [c_1, c_2, \ldots, c_K]^T$ the matrix composed by the center of the cluster. We denote $\Lambda = \min(\lambda(C), \lambda(X))$ for simplification.

We are now ready to present our main theorem that establishes the convergence of the proposed self-distillation algorithm to the *ground truth labels* under certain conditions.

**Theorem 1.** *Assume that $|\phi(0)|$, $|\phi'(\cdot)|$ and $|\phi''(\cdot)|$ are bounded with upper bound $\Gamma \geq 1$. We fix a learning rate $\eta = \frac{1}{2\Gamma^2 n}$ for the gradient descent. Assume that the sequence $\alpha_t$ monotonically decreases to $0$. Furthermore, we have two slow-decay conditions on $\alpha_t$*

- $\max\limits_{t < T_2} 2\sqrt{n}(\alpha_t - \alpha_{t+1}) \leq \frac{c_{low}\lambda(C)}{512\Gamma^2 K}(1 - 2\rho), \quad \max\limits_{s \geq T_2} 2\sqrt{n}(\alpha_s - \alpha_{s+1}) \leq \frac{c_{low}\Lambda}{512\Gamma^2 K}(1 - 2\rho),$

- $\alpha_{T_1} \geq \max(1 - \frac{c_{low}\lambda(C)}{128\Gamma^2 K}(1 - 2\rho), \frac{\frac{7}{4} - \frac{3}{2}\rho}{2 - 2\rho}),$

*where $T_1 = \lceil \frac{80\Gamma^2 K}{c_{low}\lambda(C)} \log(\frac{\Gamma\sqrt{32n \log \frac{8}{\delta}}}{1 - 2\rho}) \rceil$ and $T_2 = \inf\{t : \alpha_t < \frac{1}{24\sqrt{n}}\}$. We choose the following label function $h(\cdot)$*

$$h(x) = \begin{cases} \dfrac{4}{1 - 2\rho}x, & |x| \leq \dfrac{1 - 2\rho}{4}, \\[2mm] sgn\,(x), & |x| > \dfrac{1 - 2\rho}{4}. \end{cases}$$

*For the self-distillation algorithm, if the following two conditions for the radius $\epsilon$ and the width $k$ are satisfied*

$$\epsilon = O\left(\frac{(1 - 2\rho)^2}{\sqrt{d}nT_2^2 \log \frac{1}{\delta}}\right), \quad k = \Omega\left(\max\left\{\frac{K^3}{c_{low}^3 \Lambda^3} \log \frac{1}{\delta}, \frac{nT_2 K}{c_{low}\Lambda}, \frac{n^3 T_2^4}{(1 - 2\rho)^2} \log \frac{1}{\delta}, \frac{n}{\Lambda} \log \frac{n}{\delta}\right\}\right),$$

*then for random initialization $W_0 \sim \mathcal{N}(0, 1)^{k \times d}$, with probability $1 - \delta$, we have:*

$$\lim_{t \to \infty} \|f(W_t, X) - \tilde{y}\|_2 = 0,$$

*where $W_t$ is the parameter generated by the full-batch self-distillation algorithm at iteration $t$.*

**Theorem 2.** *Combining Theorem 1 and Neyshabur et al. (2018), with failure probability $\delta \in (0, 1)$, using the self-distillation algorithm and neural network described in Theorem 1 and under the same condition, we have*

$$\mathbb{E}_{(x,y) \sim P} \| \lim_{t \to \infty} f(W_t, x) - y \|_2^2 \leq O((1 + \frac{K\Gamma^2}{c_{low}\Lambda})\sqrt{\frac{\log \frac{1}{\delta}}{n}}).$$

Compared to previous result on noisy label Li et al. (2019), our results made the following improvements. Firstly, our algorithm fits the ground truth labels without the help of early stopping as long as a mild condition on $\alpha_t$ is satisfied. Secondly, while previous work only ensures the algorithm to yield correct class labels on training set, our results state the $\ell_2$ convergence of the outputs to the ground truth labels. As a result, the solution that our algorithm finds tends to have larger margin which leads to better generalization Mohri et al. (2018). This is shown in Theorem 2. and will be supported by our empirical studies in the next subsection.

### 3.4 NOISY LABEL REFINERY

In this section, we conduct experiments on the self-distillation algorithm. In the experiments, we applied our algorithm on corrupted Fashion MNIST and CIFAR10. At noise level $p$, every data in the original training dataset is chosen and assigned a symmetric noisy label with probability $p$. We test our algorithm for $p = 0.2, 0.4, 0.6$ and $0.8$. The test accuracy is calculated with respect to the ground truth labels.

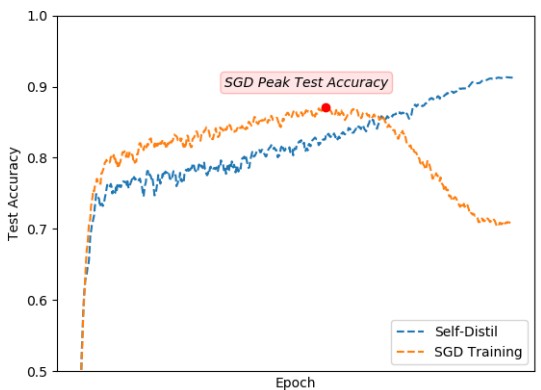

We adopted the shake-shake network of Gastaldi (2017) with 32 channels and cross entropy loss. We trained the network by momentum stochastic gradient descent (SGD) with batch size 128, momentum 0.9 and a weight decay of 1e-4. We schedule the learning rate following cosine learning rate Loshchilov & Hutter (2017) with a maximum learning rate 0.2 and minimum learning rate 0. In order to ensure convergence, we trained the models for 600 epochs. Following Lee et al. (2015), mean subtraction, hor-

Figure 4: Training on CIFAR10 with 40% noise injection. The normal training suffers from over-fitting, while self-distillation does not. (Note that we are conducting cosine learning rate scheduling. The learning rate is extremely small at the end of learning.)

izontal random flip, $32 \times 32$ random crops after padding with 4 pixels on each side from the padded image is performed as the data augmentation process. For testing, We only do evaluation on the original $32 \times 32$ image. In self-distillation, we adaptively adjust $\alpha_t$ by setting $1 - \alpha_t = \lambda * accuracy$, where $accuracy$ is the accuracy calculated on the current batch. The direct ratio $\lambda$ is the only tuning parameter. For CIFAR10, We simply set $\lambda$ as 1 when the noisy level is low, such as $p = 0, 0.2$ and 0.4. When the noisy level $p = 0.6$ and 0.8, we take $\lambda = 1.5$. For Fashion MNIST, We simply set $\lambda$ as 0.6, 1, 1, 1.4, 1.6 respectively when $p = 0, 0.2, 0.4, 0.6, 0.8$. We report the average final test accuracy of 3 runs for Fashion MNIST and CIFAR10 in Table 1.

| | CIFAR10 | | | | | Fashion MNIST | | | | |
|---|---|---|---|---|---|---|---|---|---|---|
| Noise Rate | 0 | 0.2 | 0.4 | 0.6 | 0.8 | 0 | 0.2 | 0.4 | 0.6 | 0.8 |
| CoTeachingHan et al. (2018) | 90.12 | 86.19 | 80.87 | - | - | 94.28 | 91.24 | 86.83 | - | - |
| D2LMa et al. (2018) | 91.29 | 86.64 | 73.12 | - | - | 94.47 | 89.12 | 78.98 | - | - |
| WATDamodaran et al. (2019) | 91.88 | 89.12 | 84.55 | - | - | 94.70 | 93.37 | 90.41 | - | - |
| GCEZhang & Sabuncu (2018) | - | 89.7 | 87.62 | 82.70 | 67.92 | - | 93.21 | 92.60 | 91.56 | 88.33 |
| Ours | **94.31** | **93.75** | **90.82** | **86.72** | **73.91** | **95.21** | **94.58** | **93.78** | **92.63** | **89.21** |

Table 1: Results of Self-distillation.

## 3.5 DISTILLATION $\gtrless$ EARLY STOPPING

Distillation can benefit from more than just early stopping. Distillation has the ability to enhance the AIR and thus can extract more dark knowledge from the data via early stopping. For example, Hinton et al. (2015) enhanced AIR by adjusting the temperature in the softmax layer. In self-distillation, the label function $h(\cdot)$ is proposed to amplify the information gained in the earlier epochs so that the knowledge gained in the earlier epochs is preserved. Thus, self-distillation does not require early stopping which makes the algorithm more user-friendly. On the other hand, the self-distillation algorithm dynamically enhances AIR which enables it to achieve $\ell_2$ convergence and thus better generalization. Again, we use the ratio of the norm of the label

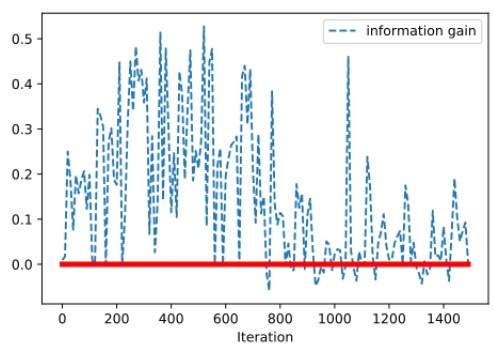

Figure 5: Self-distillation always gains information.

vector which lies in the top-5 eigenspace as a representative of informative information. We call the subtraction of informative information corresponding to the label vector provided by self-distillation algorithm and original label vector as information gain. From 5 we can see that the information gain is mostly larger than zero during the training of self-distillation algorithm. This phenomenon indicates that the supervision signal of self-distillation algorithm gains more information than directly using the noisy label. To sum up, a well-designed distillation algorithm can enjoy a regularization effect beyond early stopping and is able to gain more knowledge from the data.

## 4 CONCLUSION AND DISCUSSION

This paper provided an understanding of distillation using overparameterized neural networks. We observed that such neural networks posses the property of Anisotropic Information Retrieval (AIR), which means the neural network tends to fit the infomrative information (*i.e.* the eigenspaces associated with the largest few eigenvalues of NTK) first and the non-informative information later. Through AIR, we further observed that distillation of the Dark Knowledge is mainly due to early stopping. Based on this new understanding, we proposed a new self-distillation algorithm for noisy label refinery. Both theoretical and empirical justifications of the performance of the new algorithm were provided.

Our analysis is based on the assumption that the teacher neural network is overparameterized. When the teacher network is not overparameterized, the network will be biased towards the label even without early stopping. It is still an interesting and unclear problem that whether the bias can provide us with more information. For label refinery, our analysis is mostly based on the symmetric noise setting. We are interested in extending our analysis to the asymmetric setting.

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

## A    Proof Details

### A.1    Neural Network Properties

As preliminaries, we first discuss some properties of the neural network. We begin with the jacobian of the one layer neural network $x \to v^{\mathsf{T}}\phi(Wx)$, the Jacobian matrix with respect to $W$ takes the form

$$\mathbf{J}^{\mathsf{T}}(W) = (\mathrm{diag}(v)\phi'(WX^{\mathsf{T}})) * X^{\mathsf{T}}$$

Thus

$$\mathbf{J}(W)\mathbf{J}(W)^{\mathsf{T}} = (\phi'(XW^{\mathsf{T}})\mathrm{diag}(v)\mathrm{diag}(v)\phi'(XW^{\mathsf{T}})) \odot (XX^{\mathsf{T}})$$

First we borrow Lemma 6.6, 6.7, 6.8 from Oymak & Soltanolkotabi (2018) and Theorem 6.7, 6.8 from Li et al. (2019).

**Lemma 1.** *Let $X = [x_1, x_2, \ldots, x_n]^T$ be a data matrix made up of data with unit Euclidean norm. Assuming that $\lambda(X) > 0$, the following properties hold.*

- $\|\boldsymbol{J}(W, X) - \boldsymbol{J}(\tilde{W}, X)\| \leq \frac{\Gamma\sqrt{n}}{\sqrt{k}}\|W - \tilde{W}\|_F,$

- $\|\boldsymbol{J}(W, X)\| \leq \Gamma\sqrt{n},$

- *As long as $k \geq \frac{20\Gamma^2 n \log \frac{n}{\delta}}{\lambda(X)}$, at random Gaussian initialization $W_0 \sim \mathcal{N}(0,1)^{k \times d}$, with probability at least $1 - \delta$, we have*

$$\sigma_{min}(\boldsymbol{J}(W_0, X)) \geq \sqrt{\frac{\lambda(X)}{2}}. \tag{1}$$

**Lemma 2.** *Let $X = [x_1, x_2, \ldots, x_n]^T$ be the data matrix of a $\epsilon$-clusterable dataset. Set $\tilde{X} = [\tilde{x}_1, \tilde{x}_2, \ldots, \tilde{x}_n]^T$ in which $\tilde{x}_i$ corresponds to the center of cluster including $x_i$. What's more, we define the matrix of cluster center $C = [c_1, c_2, \ldots, c_K]^T$. Assuming that $\lambda(C) > 0$, the following properties hold.*

- $\|\boldsymbol{J}(W, \tilde{X}) - \boldsymbol{J}(\tilde{W}, \tilde{X})\| \leq \frac{\Gamma\sqrt{c_{up}n}}{\sqrt{k}}\|W - \tilde{W}\|_F \leq \frac{\Gamma\sqrt{n}}{\sqrt{k}}\|W - \tilde{W}\|_F,$

- $\|\boldsymbol{J}(W, X)\| \leq \Gamma\sqrt{c_{up}n} \leq \Gamma\sqrt{n},$

- *As long as $k \geq \frac{20\Gamma^2 K \log \frac{K}{\delta}}{\lambda(C)}$, at random Gaussian initialization $W_0 \sim \mathcal{N}(0,1)^{k \times d}$, with probability at least $1 - \delta$, we have*

$$\sigma_{min}(\boldsymbol{J}(W_0, X), \mathcal{S}_+) \geq \sqrt{\frac{c_{low}n\lambda(C)}{2K}}, \tag{2}$$

- $range(\boldsymbol{J}(W, \tilde{X})) \subset \mathcal{S}_+$ *for any parameter matrix $W$.*

Then, we gives out the perturbation analysis of the Jacobian matrix.

**Lemma 3.** *Let $X$ be a $\epsilon$-clusterable data matrix with its center matrix $\tilde{X}$. For parameter matrices $W, \tilde{W}$, we have*

$$\|\boldsymbol{J}(W, X) - \boldsymbol{J}(\tilde{W}, \tilde{X})\| \leq \frac{\Gamma\sqrt{n}}{\sqrt{k}}(\|W - \tilde{W}\|_F + \|\tilde{W}\|\epsilon + \sqrt{k}\epsilon) \tag{3}$$

*Proof.* We bound $\|\mathbf{J}(W, X) - \mathbf{J}(\tilde{W}, \tilde{X})\|$ by

$$\|\mathbf{J}(W, X) - \mathbf{J}(\tilde{W}, \tilde{X})\| \leq \|\mathbf{J}(W, X) - \mathbf{J}(\tilde{W}, X)\| + \|\mathbf{J}(\tilde{W}, X) - \mathbf{J}(\tilde{W}, \tilde{X})\|$$

The first term is bounded by Lemma 1. As to the second term, we bound it by

$$
\begin{aligned}
\|\mathbf{J}(\tilde{W}, X) - \mathbf{J}(\tilde{W}, \tilde{X})\| = & \frac{1}{\sqrt{k}} \|\phi'(\tilde{W} X^T) * X^T - \phi'(\tilde{W} \tilde{X}^T) * \tilde{X}^T\| \\
\leq & \frac{1}{\sqrt{k}} (\|\phi'(\tilde{W} X^T) * X^T - \phi'(\tilde{W} \tilde{X}^T) * X^T\| \\
& + \|\phi'(\tilde{W} \tilde{X}^T) * X^T - \phi'(\tilde{W} \tilde{X}^T) * \tilde{X}^T\|) \\
\leq & \frac{1}{\sqrt{k}} (\|\phi'(\tilde{W} X^T) - \phi'(\tilde{W} \tilde{X}^T)\| \\
& + \|\phi'(\tilde{W} \tilde{X}^T) * (X - \tilde{X})^T\|) \\
\leq & \frac{1}{\sqrt{k}} (\Gamma \|\tilde{W}\| \|X - \tilde{X}\| + \Gamma \sqrt{k} \|X - \tilde{X}\|) \\
\leq & \frac{\sqrt{n} \Gamma \epsilon}{\sqrt{k}} (\|\tilde{W}\| + \sqrt{k})
\end{aligned}
$$

Combining the inequality above, we get

$$
\|\mathbf{J}(W, X) - \mathbf{J}(\tilde{W}, \tilde{X})\| \leq \frac{\Gamma \sqrt{n}}{\sqrt{k}} (\|W - \tilde{W}\|_F + \|\tilde{W}\| \epsilon + \sqrt{k} \epsilon)
$$

$\square$

**Lemma 4.** *Let $X$ be a $\epsilon$-clusterable data matrix with its center matrix $\tilde{X}$. We assume $\|\tilde{W}_1\|, \|\tilde{W}_2\|$ have a upper bound $c\sqrt{k}$. Then for parameter matrices $W_1, W_2, \tilde{W}_1, \tilde{W}_2$, we have*

$$
\|\mathbf{J}(W_1, W_2, X) - \mathbf{J}(\tilde{W}_1, \tilde{W}_2, \tilde{X})\| \leq \Gamma \sqrt{n} (\frac{\|W_1 - \tilde{W}_1\|_F + \|W_2 - \tilde{W}_2\|_F}{2\sqrt{k}} + (c+1)\epsilon) \quad (4)
$$

*Proof.* By the definition of average Jacobian, we have

$$
\begin{aligned}
& \|\mathbf{J}(W_1, W_2, X) - \mathbf{J}(\tilde{W}_1, \tilde{W}_2, \tilde{X})\| \\
\leq & \int_0^1 \|\mathbf{J}(W_2 + \alpha(W_1 - W_2), X) - \mathbf{J}(\tilde{W}_2 + \alpha(\tilde{W}_1 - \tilde{W}_2), \tilde{X})\| d\alpha \\
\leq & \int_0^1 \frac{\Gamma \sqrt{n}}{\sqrt{k}} (\|\alpha(W_1 - \tilde{W}_1) + (1 - \alpha)(W_2 - \tilde{W}_2)\|_F + \|\tilde{W}_2 + \alpha(\tilde{W}_1 - \tilde{W}_2)\| \epsilon + \sqrt{k} \epsilon) d\alpha \\
\leq & \int_0^1 \frac{\Gamma \sqrt{n}}{\sqrt{k}} (\alpha \|W_1 - \tilde{W}_1\|_F + (1 - \alpha) \|W_2 - \tilde{W}_2\|_F + (\alpha \|\tilde{W}_1\| + (1 - \alpha) \|\tilde{W}_2\|) \epsilon + \sqrt{k} \epsilon) d\alpha \\
\leq & \Gamma \sqrt{n} (\frac{\|W_1 - \tilde{W}_1\|_F + \|W_2 - \tilde{W}_2\|_F}{2\sqrt{k}} + (c+1)\epsilon)
\end{aligned}
$$

$\square$

## A.2 PROVE OF THE THEOREM

First, we introduce the proof idea of our theorem. Our proof of the theorem divides the learning process into two stages. During the first stage, we aim to prove that the neural network will give out the right classification, *i.e.* the 0-1-loss converges to 0. The proof in this part is modified from Li et al. (2019). Furthermore, we proved that training 0-1-loss will keep 0 until the second stage starts and the margin at the first stage will larger than $\frac{1-2\rho}{2}$. During the second stage, we prove that the neural networks start to further enlarge the margin and finally the $\ell_2$ loss starts to converge to zero.

Following Oymak & Soltanolkotabi (2018); Li et al. (2019); Du et al. (2018b), we directly analysis the dynamics of each individual prediction $f(W, x_i)$ for $i = 1, 2, \cdots, n$. Oymak & Soltanolkotabi (2018); Li et al. (2019) has shown that this dynamic can be illustrated by the average Jacobian.

**Definition 3.** *We define the average Jacobian for two parameters $W_1$ and $W_2$ and data matrix $X$ as*

$$
\mathbf{J}(W_1, W_2, X) = \int_0^1 \mathbf{J}(W_2 + \alpha(W_1 - W_2), X) d\alpha. \quad (5)
$$

**Lemma 5.** *(Li et al. (2019) Lemma 6.2) Given gradient descent iterate $\hat{\theta} = \theta - \eta \nabla L(\theta)$, define*

$$C(\theta) = \boldsymbol{J}(\hat{\theta}, \theta)\boldsymbol{J}^{\mathsf{T}}(\theta)$$

*The residual $\hat{r} = f(\hat{\theta}) - y, r = f(\theta) - y$ obey the following equation*

$$\hat{r} = (I - \eta C(\theta))r$$

In our proof, we project the residual to the following subspace

**Definition 4.** *Let $\{x_i\}_{i=1}^n$ be a $\epsilon$-clusterable dataset and $\{\tilde{x}_i\}_{i=1}^n$ be the associated cluster centers, that is, $\tilde{x}_i = c_l$ iff $x_i$ is from lth cluster. We define the support subspace $\mathcal{S}_+$ as a subspace of dimension $K$, dictated by the cluster membership as follows. Let $\Lambda_l \subset \{1, 2, \cdots, n\}$ be the set of coordinates $i$ such that $= c_l$. Then $\mathcal{S}_+$ is characterized by*

$$\mathcal{S}_+ = \{v \in \mathbb{R}^n | v_{i_1} = v_{i_2} \forall i_1, i_2 \in \Lambda_l, 1 \leq l \leq K\}$$

**Definition 5.** *We define the minimum eigenvalue of a matrix $B$ on a subspace $\mathcal{S}$*

$$\sigma_{min}(B, \mathcal{S}) = \min_{\|v\|_2 = 1, UU^T = P_{\mathcal{S}}} \|v^T U^T B\|_2,$$

*where $P_{\mathcal{S}}$ is the projection to the space $\mathcal{S}$.*

Recall the generation process of the dataset

**Definition 6.** *(Clusterable Dataset Descriptions)*

- *We assume that $\{x_i\}_{i \in [n]}$ contains points with unit Euclidean norm and has $K$ clusters. Let $n_l$ be the number of points in the lth cluster. Assume that number of data in each cluster is balanced in the sense that $n_l \geq c_{low} \frac{n}{K}$ for constant $c_{low} > 0$.*

- *For each of the $K$ clusters, we assume that all the input data lie within the Euclidean ball $\mathcal{B}(c_l, \epsilon)$, where $c_l$ is the center with unit Euclidean norm and $\epsilon > 0$ is the radius.*

- *A dataset satisfying the above assumptions is called an $\epsilon$-clusterable dataset.*

### A.2.1 THE FIRST STAGE: FITTING THE LABEL

First, we reduct the dataset to its cluster center, *i.e.* $\epsilon = 0$ for $\epsilon$-clusterable dataset.

**Lemma 6.** *We fix the label function*

$$h(x) = \begin{cases} \dfrac{4}{1 - 2\rho}x & |x| \leq \dfrac{1 - 2\rho}{4} \\ sgn\,(x) & |x| > \dfrac{1 - 2\rho}{4} \end{cases}$$

*Let $\{x_i\}_{i=1}^n$ be a $\epsilon$-clusterable dataset and $\{\tilde{x}_i\}_{i=1}^n$ be the associated cluster centers, that is, $\tilde{x}_i = c_l$ iff $x_i$ is from lth cluster. We denote the data matrix $X$ and $\tilde{X}$. We denote $\alpha = \sqrt{\frac{c_{low} n \lambda(C)}{8K}}, \beta = \Gamma\sqrt{n}$ and $L = \frac{\Gamma\sqrt{n}}{\sqrt{k}}$. We set the learning rate $\eta = \min(\frac{1}{2\beta^2}, \frac{\alpha}{L\beta\Theta})$, where $\Theta$ is maximum of the residual norm during the optimization. We suppose along the optimization path we have $\alpha \leq \|\boldsymbol{J}(W, \tilde{X})v\| \leq \beta$ for all $v \in \mathcal{S}_+$. We set $T_1 = \lceil \log_{1 - \frac{n\alpha^2}{4}} \frac{1 - 2\rho}{8\|\bar{r}_0\|_2} \rceil$, where $\bar{r}_0 = P_{\mathcal{S}_+}(f(W_0, \tilde{X}) - y_0)$ is the projected residual on the space $\mathcal{S}_+$. Then $\forall t \geq T_1$, we have*

- *The neural network can learn the true label*

$$sgn\,(f)\,(W_t, \tilde{X}) = \tilde{y}.$$

- *The weight vector will be close to its initialization for all iterations*

$$\sum_{t=0}^{\infty} \|W_{t+1} - W_t\|_F \leq \eta\beta \sum_{t=0}^{\infty} \|\bar{r}_t\|_2 \leq \frac{\beta}{\alpha^2}(4\|\bar{r}_0\|_2 + 8\sqrt{n}).$$

*Proof.* We denote $C_1 \triangleq \max\limits_{t \geq 0} 2\sqrt{n}(\alpha_t - \alpha_{t+1})$. We denote $\mathbf{J}_t = \mathbf{J}(W_t, \tilde{X})$. Follow the step of gradient descent, we have

$$\|W_{t+1} - W_t\|_F = \eta\|\mathbf{J}_t^T r_t\|_2 \leq \eta\|\mathbf{J}_t^T \bar{r}_t\|_2 \leq \eta\beta\|\bar{r}_t\|_2 \tag{6}$$

We denote $\mathbf{G}_t = \mathbf{J}(W_{t+1}, W_t, \tilde{X})\mathbf{J}(W_t, \tilde{X})^T$. Then the dynamic of gradient descent can be written by

$$f(W_{t+1}, \tilde{X}) = f(W_t, \tilde{X}) - \eta\mathbf{G}_t r_t. \tag{7}$$

We consider the dynamic of the residual $r_t$

$$r_{t+1} = (I - \eta\mathbf{G}_t)r_t + y_t - y_{t+1}.$$

We project the residual on $\mathcal{S}_+$

$$\bar{r}_{t+1} = (I - \eta\mathbf{G}_t)\bar{r}_t + \bar{y}_t - \bar{y}_{t+1}.$$

Thus, the norm of the residual can be bounded by

$$\|\bar{r}_{t+1}\|_2 \leq \|(I - \eta\mathbf{G}_t)\bar{r}_t\|_2 + (1 - \alpha_t)\|h(f(W_t, \tilde{X})) - h(f(W_{t+1}, \tilde{X}))\|_2 \tag{8}$$

$$+ (\alpha_t - \alpha_{t+1})\|\bar{y} - h(f(W_{t+1}, \tilde{X}))\|_2 \tag{9}$$

$$\leq \|(I - \eta\mathbf{G}_t)\bar{r}_t\|_2 + (1 - \alpha_t)\eta L_h\beta^2\|\bar{r}_t\|_2 + 2\sqrt{n}(\alpha_t - \alpha_{t+1}) \tag{10}$$

Utilizing the following lemma

**Lemma 7.** *(Claim2. Li et al. (2019)) Let $\mathbf{P}_{\mathcal{S}_+}$ be the projection matrix to $\mathcal{S}_+$, then the following inequality holds*

$$\beta^2\mathbf{P}_{\mathcal{S}_+} \succeq \mathbf{G}_t \succeq \frac{1}{2}J_tJ_t^T \succeq \frac{\alpha^2}{2}\mathbf{P}_{\mathcal{S}_+}$$

*on the condition that $\eta \leq \frac{\alpha}{L\beta\|r_t\|_2}$.*

Therefore, as long as $\eta \leq \frac{\alpha}{L\beta\|r_t\|_2}$ holds, we have

$$\|\bar{r}_{t+1}\|_2 \leq (1 - \frac{\eta\alpha^2}{2})\|\bar{r}_t\|_2 + (1 - \alpha_t)\eta L_h\beta^2\|\bar{r}_t\|_2 + 2\sqrt{n}(\alpha_t - \alpha_{t+1}). \tag{11}$$

When $\alpha_t \geq 1 - \frac{\alpha^2}{4\beta^2 L_h}$, we have

$$\|\bar{r}_{t+1}\|_2 \leq (1 - \frac{\eta\alpha^2}{4})\|\bar{r}_t\|_2 + C_1.$$

By simple calculation, we have

$$\|\bar{r}_t\|_2 \leq (1 - \frac{\eta\alpha^2}{4})^t\|\bar{r}_0\|_2 + \frac{4}{\eta\alpha^2}C_1.$$

After $T_1 = \log_{1 - \frac{\eta\alpha^2}{4}} \frac{1 - 2\rho}{8\|\bar{r}_0\|_2}$ iterations, we have

$$\|\bar{r}_{T_1}\|_2 \leq \frac{1 - 2\rho}{4}$$

as long as $C_1 \leq \frac{\eta\alpha^2}{32}(1 - 2\rho)$. On the condition that $\alpha_{T_1} \geq 1 - \frac{\frac{7}{4} - \frac{3}{2}\rho}{2 - 2\rho}$, we have

$$\bar{y}_{T_1}(i)\tilde{y}(i) \geq \frac{3}{4}(1 - 2\rho) \quad i = 1, 2, \ldots n.$$

As a result, all the data have been classified correctly at iteration $T_1$ and the following inequality holds

$$f(W_{T_1}, \tilde{x}_i)\tilde{y}(i) \geq \frac{1 - 2\rho}{2}, \quad i = 1, 2, \ldots n.$$

For the next step, we use induction to prove that

$$h(f_t) = \tilde{y}, \|\bar{r}_t\|_2 \leq \frac{1 - 2\rho}{4}, \quad t \geq T_1. \tag{12}$$

For $t = T_1$, it has already been proved. We assume the claim is correct for arbitrary $t$, we establish the induction for $t + 1$. By applying projection to $\mathcal{S}_+$ on equation 7, we have

$$f(W_{t+1}, \tilde{X}) - \bar{y}_t = (I - \eta G_t)\bar{r}_t.$$

By applying lemma 7, we have

$$\|f(W_{t+1}, \tilde{X}) - \bar{y}_t\|_2 \leq \|\bar{r}_t\|_2 \leq \frac{1 - 2\rho}{4}.$$

Given that $h(f_t) = \tilde{y}$, we have

$$\bar{y}_t(i)\tilde{y}(i) \geq 1 - 2\rho, \quad i = 1, 2, \ldots, n.$$

Thus, for $f_{t+1}$ we have

$$f(W_{t+1}, \tilde{x}_i)\tilde{y}_i \leq \frac{3}{4}(1 - 2\rho), \quad i = 1, 2, \ldots, n.$$

By the definition of $h(\cdot)$, we deduce that $h(f_{t+1}) = \tilde{y}$. Back to equation 8, we have

$$\|\bar{r}_{t+1}\|_2 \leq \|(I - \eta\mathbf{G}_t)\bar{r}_t\|_2 + (1 - \alpha_t)\|h(f(W_t, \tilde{X})) - h(f(W_{t+1}, \tilde{X}))\|_2 \tag{13}$$
$$+ (\alpha_t - \alpha_{t+1})\|\bar{y} - h(f(W_{t+1}, \tilde{X}))\|_2 \tag{14}$$
$$\leq (1 - \frac{\eta\alpha^2}{2})\|\bar{r}_t\|_2 + 2\sqrt{n}(\alpha_t - \alpha_{t+1}) \tag{15}$$
$$\leq (1 - \frac{\eta\alpha^2}{2})\|\bar{r}_t\|_2 + C_1 \tag{16}$$
$$\leq (1 - \frac{\eta\alpha^2}{2})\frac{1 - 2\rho}{4} + \frac{\eta\alpha^2}{32}(1 - 2\rho) \tag{17}$$
$$\leq \frac{1 - 2\rho}{4} \tag{18}$$

Finally, we estimate the total variation of the parameter. Combining equation 11 and 12, we can conclude that inequality

$$\|\bar{r}_{t+1}\|_2 \leq (1 - \frac{\eta\alpha^2}{4})\|\bar{r}_t\|_2 + 2\sqrt{n}(\alpha_t - \alpha_{t+1})$$

holds for all $t \geq 0$. After taking sum on both sides for $t = 0, 1, 2, \ldots$, we have

$$\sum_{t=1}^{\infty} \|\bar{r}_t\|_2 \leq (1 - \frac{\eta\alpha^2}{4})\sum_{t=0}^{\infty} \|\bar{r}_t\|_2 + 2\sqrt{n}.$$

By simple calculation, we get

$$\sum_{t=0}^{\infty} \|\bar{r}_t\|_2 \leq \frac{4\|\bar{r}_0\|_2 + 8\sqrt{n}}{\eta\alpha^2}.$$

Combining equation 6, we have

$$\sum_{t=0}^{\infty} \|W_{t+1} - W_t\|_F \leq \eta\beta \sum_{t=0}^{\infty} \|\bar{r}_t\|_2 \leq \frac{\beta}{\alpha^2}(4\|\bar{r}_0\|_2 + 8\sqrt{n}).$$

$\square$

### A.2.2 SECOND STAGE: ENLARGE THE MARGIN

Then we further to adapt the above theorem to the $\epsilon$-clusterable dataset by a pertubation analysis. Since we have a simple inequality $\alpha_a \geq \alpha_b$, in the following discussion, we simply replace the $\alpha_a$ in the previous conclusion by $\alpha_b$, the conclusion also holds.

**Lemma 8.** *Let $\{x_i\}_{i=1}^n$ be a $\epsilon$-clusterable dataset and $\{\tilde{x}_i\}_{i=1}^n$ be the associated cluster centers, that is, $\tilde{x}_i = c_l$ iff $x_i$ is from lth cluster. We denote the data matrix $X$ and $\tilde{X}$. For the same initialization $W_0 = \tilde{W}_0$, we run the self-distillation algorithm on $X$ and $tX$ respectively. We denote the parameter matrix $W_t$ and $\tilde{W}_t$ for $t \geq 0$. We denote $\alpha = \sqrt{\frac{c_{low} n \Lambda}{8K}}, \beta = \Gamma \sqrt{n}$ and $L = \frac{\Gamma \sqrt{n}}{\sqrt{k}}$. We set the learning rate $\eta = \min(\frac{1}{2\beta^2}, \frac{\alpha}{L\beta\Theta})$, where $\Theta$ is maximum of the residual norm during the optimization. We denote $c\sqrt{k}$ the upper bound of the Frobenius norm of the parameter matrxi and we set $M = c + 1$. We set $T_2 = \inf\{t : \alpha_t < \frac{1}{24\sqrt{n}}\}$. Then if the following conditions hold*

$$\epsilon \leq \frac{1 - 2\rho}{4M\eta\beta(2 + L_h)\Gamma\sqrt{n}T_2(1 + \Theta)}$$
$$k \geq 4\eta^2\Gamma^2\Theta^2 n(2\beta^2(2 + L_h)^2 T_2 + 1)^2 T_2^2,$$

*we have*

$$\|f(W_t, X) - f(\tilde{W}_t, \tilde{X})\|_2 \leq 4\eta\beta(2 + L_h)\Theta M\Gamma\sqrt{n}\epsilon t$$
$$\|W_t - \tilde{W}_t\|_F \leq 2\eta M\Gamma\Theta\sqrt{n}(2\beta^2(2 + L_h)^2 T_2 + 1)\epsilon t$$

*for all $t \leq T_2$.*

*Proof.* We introduce the following notations.

$$r_t = f(W_t, X) - y_t, \mathrm{Tr}_t = f(\tilde{W}_t, X) - \tilde{y}_t$$
$$\mathbf{J}_t = \mathbf{J}_t(W_t, X), \tilde{\mathbf{J}}_t = \tilde{\mathbf{J}}_t(\tilde{W}_t, \tilde{X})$$
$$\mathbf{J}_{t+1,t} = \mathbf{J}(W_{T+1}, W_t, X), \tilde{\mathbf{J}}_{t+1,t} = \mathbf{J}(\tilde{W}_{T+1}, \tilde{W}_t, \tilde{X})$$
$$d_t = \|W_t - \tilde{W}_t\|_F, p_t = \|f(W_t, X) - f(\tilde{W}_t, \tilde{X})\|_2.$$

We can conclude the following inequalities from lemma 3 and lemma 4

$$\|\mathbf{J}_t - \tilde{\mathbf{J}}_t\| \leq Ld_t + M\Gamma\sqrt{n}\epsilon,$$
$$\|\mathbf{J}_{t+1,t} - \tilde{\mathbf{J}}_{t+1,t}\| \leq L\frac{d_t + d_{t+1}}{2} + M\Gamma\sqrt{n}\epsilon.$$

Thus the parameters are updated by gradient descent, we have

$$\begin{aligned}
d_{t+1} = \|W_{t+1} - \tilde{W}_{t+1}\|_F &\leq \|W_t - \tilde{W}_t\|_F + \|\eta\mathbf{J}_t^T r_t - \eta\tilde{\mathbf{J}}_t^T \mathrm{Tr}_t\|_2 \\
&\leq d_t + \eta\|\mathbf{J}_t - \tilde{\mathbf{J}}_t\|\|\mathrm{Tr}_t\|_2 + \eta\|\mathbf{J}_t\|\|r_t - \mathrm{Tr}_t\|_2 \\
&\leq d_t + \eta(L\Theta d_t + M\Gamma\Theta\sqrt{n}\epsilon + \beta(1 + L_h)p_t)
\end{aligned}$$

Also, we have

$$\begin{aligned}
p_{t+1} =&\|f(W_{t+1}, X) - f(\tilde{W}_{t+1}, \tilde{X}))\|_2 \\
\leq&\|f(W_t, X) - f(\tilde{W}_t, \tilde{X}) - \eta\tilde{\mathbf{J}}_{t+1,t}\tilde{\mathbf{J}}_t^T (r_t - \mathrm{Tr}_t)\|_2 \\
&+ \eta\|(\mathbf{J}_{t+1,t} - \tilde{\mathbf{J}}_{t+1,t})\mathbf{J}_t^T r_t\|_2 + \eta\|\tilde{\mathbf{J}}_{t+1,t}(\mathbf{J}_t^T - \tilde{\mathbf{J}}_t^T)r_t\|_2 \\
\leq&\|f(W_t, X) - f(\tilde{W}_t, \tilde{X}) - \eta\tilde{\mathbf{J}}_{t+1,t}\tilde{\mathbf{J}}_t^T (r_t - \mathrm{Tr}_t)\|_2 \\
&+ \eta\beta\|r_t\|_2(L\frac{3d_t + d_{t+1}}{2} + 2M\Gamma\sqrt{n}\epsilon) \\
\leq&\|(1 - \eta\tilde{\mathbf{J}}_{t+1,t}\tilde{\mathbf{J}}_t^T)(f(W_t, X) - f(\tilde{W}_t, \tilde{X}))\|_2 + \eta\|\tilde{\mathbf{J}}_{t+1,t}\tilde{\mathbf{J}}_t^T (y_t - \tilde{y}_t)\|_2 \\
&+ \eta\beta\|r_t\|_2(L\frac{3d_t + d_{t+1}}{2} + 2M\Gamma\sqrt{n}\epsilon) \\
\leq&(1 - \frac{\eta\alpha^2}{2})p_t + \eta\beta^2(1 - \alpha_t)\|h(f(W_t, X)) - h(f(\tilde{W}_t, \tilde{X}))\|_2 \\
&+ \eta\beta\|r_t\|_2(L\frac{3d_t + d_{t+1}}{2} + 2M\Gamma\sqrt{n}\epsilon)
\end{aligned}$$

If $p_t \leq \frac{1-2\rho}{4}$ holds for $t \leq T_2$, then for $T_1 \leq t \leq T_2$, we have $f(W_t, X) = \tilde{y}$. Under such circumstance, we have $\|h(f(W_t, X)) - h(f(\tilde{W}_t, \tilde{X}))\|_2 = 0$. For $t < T_1$, we have

$$(1 - \frac{\eta\alpha^2}{2})p_t + \eta\beta^2(1 - \alpha_t)\|h(f(W_t, X)) - h(f(\tilde{W}_t, \tilde{X}))\|_2$$

$$\leq (1 - \frac{\eta\alpha^2}{2})p_t + \eta\beta^2(1 - \alpha_t)L_h p_t$$

$$\leq (1 - \frac{\eta\alpha^2}{2})p_t + \eta\beta^2\frac{\alpha^2}{4\beta^2 L_h}L_h p_t$$

$$\leq p_t.$$

To sum up, if we can guarantee that $p_t \leq \frac{1-2\rho}{4}$ holds for $t \leq T_2$, we have

$$p_{t+1} \leq p_t + \eta\beta\|r_t\|_2(L\frac{3d_t + d_{t+1}}{2} + 2M\Gamma\sqrt{n}\epsilon).$$

For $\|r_t\|_2$, we have

$$\|r_t\|_2 \leq \|\operatorname{Tr}_t\|_2 + \|r_t - \operatorname{Tr}_t\|_2$$

$$\leq \|\operatorname{Tr}_t\|_2 + (1 + L_h)\|f(W_t, X) - f(\tilde{W}_t, \tilde{X})\|_2$$

$$\leq \|\operatorname{Tr}_t\|_2 + (1 + L_h)p_t$$

$$\leq \Theta + (1 + L_h)p_t.$$

Thus we have the following inequality for $p_t$

$$p_{t+1} \leq p_t + \eta\beta(\Theta + (1 + L_h)p_t)(L\frac{3d_t + d_{t+1}}{2} + 2M\Gamma\sqrt{n}\epsilon).$$

We claim that if the following conditions for $\epsilon$ and $k$ hold

$$\epsilon \leq \frac{1 - 2\rho}{4M\eta\beta(2 + L_h)\Gamma\sqrt{n}T_2(1 + \Theta)}$$

$$k \geq 4\eta^2\Gamma^2\Theta^2 n(2\beta^2(2 + L_h)^2 T_2 + 1)^2 T_2^2,$$

one can show that

$$p_t \leq 4\eta\beta(2 + L_h)\Theta M\Gamma\sqrt{n}\epsilon t$$

$$d_t \leq 2\eta M\Gamma\Theta\sqrt{n}(2\beta^2(2 + L_h)^2 T_2 + 1)\epsilon t$$

for $t \leq T_2$ by induction. It is obviously when $t = 0$. We further suppose the inequalities hold for an arbitrary $t$ satisfying $t < T_2$, we have

$$d_{t+1} \leq d_t + \eta(L\Theta d_t + M\Gamma\Theta\sqrt{n}\epsilon + \beta(1 + L_h)p_t)$$

$$\leq d_t + \eta(2M\Gamma\Theta\sqrt{n}\epsilon + 4\eta\beta(2 + L_h)\Theta M\Gamma\sqrt{n}\epsilon t\beta(1 + L_h))$$

$$\leq d_t + \eta(2M\Gamma\Theta\sqrt{n}\epsilon + 4\eta\beta^2(2 + L_h)^2\Theta M\Gamma\sqrt{n}\epsilon T_2)$$

$$\leq 2\eta M\Gamma\Theta\sqrt{n}(2\beta^2(2 + L_h)^2 T_2 + 1)\epsilon(t + 1)$$

because of the condition on $k$ ensures that $L\Theta d_t \leq M\Gamma\Theta\sqrt{n}\epsilon$. When it comes to $p_{t+1}$, we have

$$p_{t+1} \leq p_t + \eta\beta(\Theta + (1 + L_h)p_t)(L\frac{3d_t + d_{t+1}}{2} + 2M\Gamma\sqrt{n}\epsilon)$$

$$\leq p_t + \eta\beta(\Theta + (1 + L_h)\Theta)(4L\eta M\Gamma\Theta\sqrt{n}(2\beta^2(2 + L_h)^2 T_2 + 1)\epsilon(t + 1) + 2M\Gamma\sqrt{n}\epsilon)$$

$$\leq p_t + \eta\beta(\Theta + (1 + L_h)\Theta)(2Ld_{T_2} + 2M\Gamma\sqrt{n}\epsilon)$$

$$\leq p_t + \eta\beta(2 + L_h)\Theta(2M\Gamma\sqrt{n}\epsilon + 2M\Gamma\sqrt{n}\epsilon)$$

$$\leq p_t + 4\eta\beta(2 + L_h)\Theta M\Gamma\sqrt{n}\epsilon$$

$$\leq 4\eta\beta(2 + L_h)\Theta M\Gamma\sqrt{n}\epsilon(t + 1)$$

because of the conditions on $\epsilon$ and $k$ ensure that $Ld_{T_2} \leq M\Gamma\sqrt{n}\epsilon$ and $p_t \leq \Theta$.

$\square$

Now we are ready to finalize the proof of our main theorem.

*Proof.* We denote $C_2 \triangleq \max_{s \geq T_2} 2\sqrt{n}(\alpha_s - \alpha_{s+1})$. We take $\Theta = (C_3\Gamma\sqrt{\log \frac{8}{\delta}} + 1)\sqrt{n}$. $C_3$ is the constant in Hoeffding's inequality. By lemma 6, we have $\Theta \geq \max_{t \geq 0} \|\bar{\mathrm{Tr}}_t\|_2$ with probability $1 - \frac{\delta}{4}$. Combining lemma 6 and 8, we have $f(W_t, X) = \tilde{y}$ for $T_1 \leq t \leq T_2$ and

$$
\begin{aligned}
\|r_{T_2}\|_2 &= \|f(W_{T_2}, X) - y_{T_2}\|_2 \\
&\leq \|f(W_{T_2}, X) - f(\tilde{W}_{T_2}, \tilde{X})\|_2 + \|f(\tilde{W}_{T_2}, \tilde{X}) - \bar{\tilde{y}}_{T_2}\|_2 + \|\bar{\tilde{y}}_{T_2} - y_{T_2}\|_2 \\
&\leq \|f(W_{T_2}, X) - f(\tilde{W}_{T_2}, \tilde{X})\|_2 + \|\bar{\mathrm{Tr}}_{T_2}\|_2 + \alpha_{T_2}\|\bar{y} - y\|_2 \\
&\leq \frac{1 - 2\rho}{4} + \frac{1 - 2\rho}{4} + \frac{1}{24}
\end{aligned}
$$

Similar to the proof in 6, we consider the gradient descent on original dataset $X$ after $T_2$. We proof the following claim by induction

$$
h(f(W_s), X) = \tilde{y}, \|r_s\|_2 \leq \frac{5}{8}(1 - 2\rho) + \frac{1}{24}, \quad s \geq T_2.
$$

For $s = T_2$, it has already been proved. We assume the claim is correct for arbitrary $s$, we establish the induction for $s + 1$. By equation 7, we have

$$
f(W_{s+1}, \tilde{X}) - y_s = (I - \eta G_s)r_s.
$$

By applying lemma 7, we have

$$
\|f(W_{s+1}, \tilde{X}) - \bar{y}_s\|_2 \leq \|r_s\|_2 \leq \frac{5}{8}(1 - 2\rho) + \frac{1}{24}.
$$

Given that $h(f(W_s, X)) = \tilde{y}$, we have

$$
y_s(i)\tilde{y}(i) \geq 1 - 2\alpha_{T_2}, \quad i = 1, 2, \ldots, n.
$$

Thus, for $f(W_{s+1}, x_i)$ we have

$$
f(W_{s+1}, x_i)\tilde{y}(i) \geq 1 - 2\alpha_{T_2} - \frac{5}{8}(1 - 2\rho) - \frac{1}{24} \geq \frac{1 - 2\rho}{4}.
$$

As a result, we have $f(W_{s+1}, X) = \tilde{y}$. Furthermore, we can bound $\|r_{s+1}\|_2$ by

$$
\begin{aligned}
\|r_{s+1}\|_2 \leq &\|(I - \eta\mathbf{G}_s)r_s\|_2 + (1 - \alpha_s)\|h(f(W_s, X)) - h(f(W_{s+1}, X))\|_2 \\
&+ (\alpha_s - \alpha_{s+1})\|\bar{y} - h(f(W_{s+1}, X))\|_2 \\
\leq &(1 - \frac{\eta\alpha^2}{2})\|r_s\|_2 + C_2 \\
\leq &(1 - \frac{\eta\alpha^2}{2})(\frac{5}{8}(1 - 2\rho) + \frac{1}{24}) + \frac{\eta\alpha^2}{32}(1 - 2\rho) \\
\leq &\frac{5}{8}(1 - 2\rho) + \frac{1}{24}.
\end{aligned}
$$

To sum up, we have the following inequality holds for all $s \geq T_2$

$$
\|r_{s+1}\|_2 \leq (1 - \frac{\eta\alpha^2}{2})\|r_s\|_2 + 2\sqrt{n}(\alpha_s - \alpha_{s+1}).
$$

After taking sum on both sides for $s \geq T_2$, we have

$$
\sum_{s=T_2}^{\infty} \|r_s\|_2 \leq \frac{2\|r_{T_2}\|_s + \frac{1}{6}}{\eta\alpha^2} \leq \frac{\frac{5}{4} - 2\rho}{\eta\alpha^2}.
$$

Furthermore, we have

$$
\sum_{s=T_2}^{\infty} \|W_{s+1} - W_s\|_F \leq \eta\beta \sum_{s=T_2}^{\infty} \|r_s\|_2 \leq \frac{\beta}{\alpha^2}(\frac{5}{4} - 2\rho).
$$

Finally, we check all the conditions for $\alpha$ and $\eta$.

For $\eta$, we require $\eta \leq \frac{1}{2\beta^2}$ and $\eta \leq \frac{\alpha}{L\beta} \min_{t \geq 0, s \geq T_2} \left( \frac{1}{\|\bar{\mathrm{Tr}}_t\|_2}, \frac{1}{\|r_s\|_2} \right)$. For $k \geq \frac{2n(C_3\Gamma\sqrt{\log \frac{8}{\delta}}+1)^2 K}{c_{low}\Lambda} = O(\frac{nK \log \frac{1}{\delta}}{c_{low}\Lambda})$, we have

$$\frac{\alpha}{L\beta} \min_{t \geq 0, s \geq T_2} \left( \frac{1}{\|\bar{\mathrm{Tr}}_t\|_2}, \frac{1}{\|r_s\|_2} \right) \geq \frac{1}{2\beta^2}.$$

On such condition, we can choose $\eta = \frac{1}{2\beta^2} = \frac{1}{2\Gamma^2 n}$. The distance between the intermediate parameter matrix and the initial parameter matrix can be bounded by

$$R = \max_{t \geq 0, s \geq 0}(\|W_s - W_0\|_F, \|\tilde{W}_t - \tilde{W}_0\|_F) \leq \frac{\beta}{\alpha^2}(4\|\bar{r}_0\|_2 + 8\sqrt{n}) + d_{T_2} + \frac{\beta}{\alpha^2}(\frac{5}{4} - 2\rho) \quad (19)$$

$$\leq \frac{8K\Gamma}{c_{low}\Lambda}(4C_3\Gamma\sqrt{\log \frac{8}{\delta}} + 13) + d_{T_2}. \quad (20)$$

By lemma 1 and lemma 2, as long as $k \geq \frac{20\Gamma^2 n \log \frac{4n}{\delta}}{\Lambda}$, with probability $1 - \frac{\delta}{2}$, we have

$$\sigma_{min}(\mathbf{J}(W_0, X)), \sigma_{min}(\mathbf{J}(W_0, \tilde{X}), \mathcal{S}_+) \geq 2\alpha$$

To ensure $\alpha$ lower bounding the eigenvalue of the gram matrix, we need to verify that

$$RL = R\frac{\Gamma\sqrt{n}}{\sqrt{k}} \leq \alpha$$

That is to say

$$k \geq \frac{8K\Gamma^2 R^2}{c_{low}\Lambda} \quad (21)$$

Another condition related to $R$ is the condition on $M$. We require

$$(M - 1)\sqrt{k} \geq \|W_0\|_F + R.$$

By Bernstein's inequality, we have

$$\|W_0\|_F \leq \sqrt{d + C_4 \log \frac{8}{\delta}}\sqrt{k}$$

with probability $1 - \delta/4$. On the condition that $k \geq R^2$ (acutally one can show that $\frac{8K\Gamma^2 R^2}{c_{low}\Lambda} \geq R^2$), we can choose $M = \sqrt{d + C_4 \log \frac{8}{\delta}} + 2$.

We take these constants to the lemma 8. Firstly, for $\epsilon$ we have

$$\epsilon \leq \frac{1 - 2\rho}{4M\eta\beta(2 + L_h)\Gamma\sqrt{n}T_2(1 + \Theta)}$$

$$= \frac{1 - 2\rho}{2(\sqrt{d + C_4 \log \frac{8}{\delta}} + 2)(2 + \frac{4}{1-2\rho})(1 + (C_3\Gamma\sqrt{\log \frac{8}{\delta}} + 1)\sqrt{n})}$$

$$= O(\frac{(1 - 2\rho)^2}{\sqrt{nd}T_2 \log \frac{1}{\delta}}).$$

For $k$ we have

$$k \geq 4\eta^2\Gamma^2\Theta^2 n(2\beta^2(2 + L_h)^2 T_2 + 1)^2 T_2^2$$

$$= 2\Theta^2(2\Gamma^2 n(2 + \frac{4}{1 - 2\rho})^2 T_2 + 1)^2 T_2^2$$

$$= \frac{n^3 T_2^4 \log \frac{1}{\delta}}{(1 - 2\rho)^2}$$

We point out that

$$O(\frac{n^3 T_2^4 \log \frac{1}{\delta}}{(1-2\rho)^2}) \geq O(\frac{nK \log \frac{1}{\delta}}{c_{low}\Lambda}).$$

Secondly we bound $d_{T_2}$ by

$$d_{T_2} \leq 2\eta M\Gamma\Theta\sqrt{n}(2\beta^2(2+L_h)^2 T_2 + 1)\epsilon T_2 \tag{22}$$

$$\leq \frac{(1-2\rho)2\eta M\Gamma\Theta\sqrt{n}(2\beta^2(2+L_h)^2 T_2 + 1)T_2}{4M\eta\beta(2+L_h)\Gamma\sqrt{n}T_2(1+\Theta)} \tag{23}$$

$$\leq (1-2\rho)(\beta(2+L_h)T_2 + \frac{1}{2\beta(2+L_h)}) \tag{24}$$

$$= O(\sqrt{n}T_2). \tag{25}$$

We combine equation 20, 21 and 25, we deduce the last condition for $k$

$$k = \Omega\left(\max\left\{\frac{K^3}{c_{low}^3\Lambda^3}\log\frac{1}{\delta}, \frac{nKT_2}{c_{low}\Lambda}\right\}\right)$$

To sum up, we require $k$ to satisfy

$$k = \Omega\left(\max\left\{\frac{K^3}{c_{low}^3\Lambda^3}\log\frac{1}{\delta}, \frac{nKT_2}{c_{low}\Lambda}, \frac{n^3 T_2^4}{(1-2\rho)^2}\log\frac{1}{\delta}, \frac{n}{\Lambda}\log\frac{n}{\delta}\right\}\right)$$

and require $\epsilon$ to satisfy

$$\epsilon = O\left(\frac{(1-2\rho)^2}{\sqrt{n}dT_2\log\frac{1}{\delta}}\right)$$

We substitute $C_1$ and $C_2$ by the value of $\alpha$ in lemma 6 and lemma 8 respectively and then get the condition for $\alpha_t$

- $\max_{t<T_2} 2\sqrt{n}(\alpha_t - \alpha_{t+1}) \leq \frac{c_{low}\lambda(C)}{512\Gamma^2 K}(1-2\rho), \quad \max_{s\geq T_2} 2\sqrt{n}(\alpha_s - \alpha_{s+1}) \leq \frac{c_{low}\Lambda}{512\Gamma^2 K}(1-2\rho),$

- $\alpha_{T_1} \geq \max(1 - \frac{c_{low}\lambda(C)}{128\Gamma^2 K}(1-2\rho), \frac{\frac{7}{4}-\frac{3}{2}\rho}{2-2\rho}).$

Also, we substitute $T_1$ by the value of $\alpha$ in lemma 6, and combine the fact that $\frac{\eta\alpha^2}{4} = \frac{\alpha^2}{8\beta^2} \leq \frac{1}{8}$, we have

$$T_1 = \log_{1-\frac{\eta\alpha^2}{4}}\frac{1-2\rho}{8\|\bar{r}_0\|_2}$$

$$\leq \frac{\log\frac{8\|\bar{r}_0\|_2}{1-2\rho}}{\log\frac{1}{1-\frac{\eta\alpha^2}{4}}}$$

$$\leq \frac{5}{\eta\alpha^2}\log\frac{8\|\bar{r}_0\|_2}{1-2\rho}$$

$$\leq \lceil\frac{80\Gamma^2 K}{c_{low}\lambda(C)}\log(\frac{\Gamma\sqrt{32n\log\frac{8}{\delta}}}{1-2\rho})\rceil.$$

$\square$

# B   EXPERIMENT DETIALS

**Experiment In Section2.1**   The network structure of the small student network is demonstrated in Table 2.

| Type | Kernel | Dilation | Stride | Outputs | Remark |
|---|---|---|---|---|---|
| conv. | 3×3 | 1 | 1×1 | 32 | bn |
| maxpool. | 2×2 | - | - | - | |
| conv. | 3×3 | 1 | 1×1 | 64 | bn |
| maxpool. | 2×2 | - | - | - | |
| conv. | 3×3 | 1 | 1×1 | 128 | bn |
| maxpool. | 2×2 | - | - | - | |
| Flatten | | | | | |
| fc. | - | - | - | 512 | bn |
| fc. | - | - | - | 10 | dropout |

Table 2: Architecture of the student network. After each convolution layer, there is a Rectified Linear Unit(ReLU) layer.

**Experiment In Section2.3 and Section3.5**   For these two experiments, we modify CIFAR10 to a binary classification task. We choose class $2, 7$ to be the positive class and the others to be negative. We train resnet56 with MSE loss. We set batch size $128$, momentum $0.9$, weight decay $5e - 4$ and learning rate $0.1$.

In the experiment in section2.3, we fetch a batch of data (batch size=$128$) from the testset and randomly corrupted the label by noise level $0, 0.1, 0.2, 0.3, 0.4, 0.5$. We plot the ratio of the norm of the label vector which lies in the subspaces corresponding to top-5 eigenvalues of NTK.

In the experiment in section2.3, we calculate the ratio of the norm of the label vector which lies in the subspaces corresponding to top-5 eigenvalues of NTK firstly. We calculate the ratio of the norm of the label vector provided by the self-distillation algorithm lies in the top-5 eigenspace. We calculate the difference of the latter ratio and the former ratio and called it information gain. We plot the information gain of the first 1500 iterations.

