# OpenReview forum: "Distillation $\approx$ Early Stopping? Harvesting Dark Knowledge Utilizing Anisotropic Information Retrieval For Overparameterized NN"
_ICLR.cc/2020/Conference — Reject_

### Official Review · AnonReviewer1 · 2019-10-21
**Official Blind Review #1**

**Rating:** 8

**Review:**

This paper provides a theoretical framework to understand the regularization effect of distillation. Based on the observation that a overparameterized NN has the ability to memorize all the data, thus early stopping is essential for an overparameterized teacher network to extract dark knowledge from the hard labels, the author starts to analysis distillation from an early stopping view point.

Then the author consider to use distillation to learning with corrupted labels inspired by the previous works using early stopping to clean label noise. The author also tries to formulate the “informative information” via the NTK theory. Theoretically, the author provide a proof of convergence to the ground truth labels in terms of l2 distance. While the previous result was on convergence in 0-1 loss, which means the distillation can enlarge the margin the classifier. From this proof, aurthors also make an interesting discussion to show how distillation introduces further information rather than just early stopping. Authors also demonstrate their result on Fashion MNIST and CIFAR-10 to convince the reader the benefit of the algorithm.

The paper is interesting and brings new theoretically thoughts to understand the distillation method. The relationship between early stopping and distillation can inspire the machine learning researchers to explore more about distillation both empirically and theoretically.

Minor Questions:
1. The analysis is based on the assumption that the teacher is overparameterized. What will happen if the teacher network is not overparameterized?
2. Does the assumption of dataset is too strong in the theorem?
3. Some notation is not clear, e.x. in  Algorithm1 what is $\mathcal{N}(x_i, w_t)$ and what is base network and mother network? These should be instead by student network and teacher network.
4. Author claims that early stopping is hard to tune while introducing extra hyper-parameters in the self-distillation algorithm.  Would the extra hyper-parameter makes the algorithm might be even harder to tune?

**Experience Assessment:**

I have published in this field for several years.

**Review Assessment: Checking Correctness Of Derivations And Theory:**

I carefully checked the derivations and theory.

**Review Assessment: Checking Correctness Of Experiments:**

I carefully checked the experiments.

**Review Assessment: Thoroughness In Paper Reading:**

I read the paper thoroughly.

---

> ### Author Response · Authors · 2019-11-13
> **response**
>
> We would like to thank Reviewer  for their review and helpful suggestions and here is the response from the author
> we have fixed the typos and notation problems
> For the data assumption, I think it’s just like data is sampling coming from mixtures of separable distributions. It’s a common data assumption for neural network and better than linear separable. (Li Y, Liang Y. Learning overparameterized neural networks via stochastic gradient descent on structured data[C]//Advances in Neural Information Processing Systems. 2018: 8157-8166.)
> Regarding The Hyperparameter. For early stopping, you should evaluate your model after every epoch’s update for your experiment. Also, using contaminated validation set will prevent us from selecting optimal stopping time precisely.  When it comes to the self-distillation algorithm, we can simply set $ \lambda= 1$ for moderate noise (as we mentioned in the experiments setting). Furthermore, as we claimed in the paper, self-distillation is better than early stopping because of AIR.
> Regarding the case the teacher is not overparameterized. It’s an interesting question. The reason we cannot analyze it is because the difficulty of analyzing regularization effect of underparameterized neural networks.

---

### Official Review · AnonReviewer2 · 2019-10-27
**Official Blind Review #2**

**Rating:** 3

**Review:**

This paper proposes a self-distillation algorithm for training an over-parameterized neural network with noisily labelled data. It is shown that for a binary classification task on clustered data (same as [Li et al. 2019]), even if the labels are corrupted, the self-distillation algorithm applied on a sufficiently wide two-layer network can recover the correct labels (in l_2 loss). Experiments on CIFAR-10 and Fashion MNIST are provided, which show that the self-distillation algorithm is effective for label noise with relatively little tuning.

Although the theoretical part of the paper has a large overlap with [Li et al. 2019], I find the self-distillation algorithm very interesting and it's nice that it can achieve zero l_2 loss w.r.t the correct labels. However, I think the paper could still use some improvement.

1. The theorem in Section 3.3 is only for binary classification. Can it be generalized to multi-class classification?

2. The theoretical guarantee is only for training data. Is it possible to prove a generalization bound? There's a remark on top of page 8 about margin. It would be nice to elaborate on this and maybe make it formal.

3. Section 2 is not very satisfying. I don't quite see the point of this section. In particular, the concept of AIR is nothing new and its connection to NTK top eigenspace has already been written in previous work (e.g. [Arora et al. 2019]). I'd suggest to not have this section and to make Section 3 the main contribution of this paper.


-------
update:
Thanks to the authors for the response and for adding a generalization bound.

**Experience Assessment:**

I have published one or two papers in this area.

**Review Assessment: Checking Correctness Of Derivations And Theory:**

I assessed the sensibility of the derivations and theory.

**Review Assessment: Checking Correctness Of Experiments:**

I assessed the sensibility of the experiments.

**Review Assessment: Thoroughness In Paper Reading:**

I read the paper at least twice and used my best judgement in assessing the paper.

---

> ### Author Response · Authors · 2019-11-07
> **Aurthor Response**
>
> We thank reviewer2 for his insightful comments.  Based on his comments, we want to do the following justifcation and revision of our draft
>
> > 1. The theorem in Section 3.3 is only for binary classification. Can it be generalized to multi-class classification?
>
> Yes,  but the theorem will become very nasty thus we don’t conclude it in. (We just want a theorem to reveal our idea: why distillation can clean label noise and what’s its benefit)
>
> > 2. The theoretical guarantee is only for training data. Is it possible to prove a generalization bound? There's a remark on top of page 8 about margin. It would be nice to elaborate on this and maybe make it formal.
>
> Yes, you can just easily modify a margin bound like [1]. It’s a direct modification thus we don’t include it in our first version of the draft. You can just use theorem 1.1. For we converged to the minima with a large margin, in margin bound we can let the margin gamma to the largest 1and now \hat R_\gamma = 0, we’ve added it in our modified draft.  But for [Li et al 2019] the bound \gamma can only be guaranteed as 0 thus they can’t have a generalization bound. i.e the generalization error can be trivially bounded by the norm of the neural network.
> [1] Bartlett P L, Foster D J, Telgarsky M J. Spectrally-normalized margin bounds for neural networks[C]//Advances in Neural Information Processing Systems. 2017: 6240-6249.
>
> The reason why here we need a margin is that we want to move from the estimator of the  Rademacher complexity of neural network N to the  Rademacher complexity of the loss function loss(N), we need the loss function `loss` to be lip continuous. But 0-1 loss is not lipped continuous, thus we need to utilize the margin to construct a surrogate loss.
>
>
>
> > 3. Section 2 is not very satisfying. I don't quite see the point of this section. In particular, the concept of AIR is nothing new and its connection to NTK top eigenspace has already been written in previous work (e.g. [Arora et al. 2019]). I'd suggest to not have this section and to make Section 3 the main contribution of this paper.
>
> Yes, I agree. You can go through the introduction 1.1 part to see the contribution we conclude.  I think I should separate 2.1 and another part of section2. We introduce AIR is just because we need a notation/concept we can use to formalize the idea we explain in section 3.  Section2 seems more like a “preliminary” section. At the same time, besides section3, bridging early stopping and distillation is the main contribution as mentioned by Reviewer#1. Do you have any suggestions to help us modify this section?

---

> ### Author Response · Authors · 2019-11-15
> **Update of our generalization bound**
>
> We combine the Rademacher complexity estimation in
> Behnam Neyshabur, Zhiyuan Li, Srinadh Bhojanapalli, Yann LeCun, and Nathan Srebro. Towards
> understanding the role of over-parametrization in generalization of neural networks. arXiv preprint
> arXiv:1805.12076, 2018.
> of distance from initialization ||W_t-W_0|| and further using the distance from the initialization bound in our proof to have a generalization bound without the weight of neural network in theorem2.

---

### Official Review · AnonReviewer3 · 2019-11-01
**Official Blind Review #3**

**Rating:** 1

**Review:**

This paper proposes a new perspective on understanding knowledge distillation as a transfer of information defined with respect to the neural tangent kernel. Additionally, a new framework for learning the classifier on a noisy labeled dataset is proposed based on the knowledge transfer framework.

Overall, I think the paper lacks justification (and explanation) for its main statement on how knowledge distillation is related to the early stopping of the teacher network. Especially, it is confusing since Section 2 and 3 make different statements. Specifically, Section 2 shows that early stopping "helps" knowledge distillation while Section 3 shows that knowledge distillation can "replace" early stopping. The former observation implies that early stopping is complementary to knowledge distillation, while the latter implies otherwise.

Furthermore, Section 2 mainly explains why the eigenspaces associated with the largest eigenvalue of the neural tangent kernel is "informative information". However, there is no elaboration on how the knowledge distillation process leads to the transfer of such information, i.e., there is no connection between the neural tangent kernel and the knowledge distillation process. Although Figure 1. suggests that early stopping indeed improves the knowledge distillation process, they are not enough to support the statement convincingly enough.

Without proper support on the main statement of this paper, the paper looses much of its claimed contributions. The label refinery algorithm for the noisy labeled dataset is interesting, but it is not evaluated thoroughly enough to demonstrate its superiority over existing algorithms. It also does not have much originality when compared to similar algorithms [1, 2]. Bagherinezhad  et al., [1] also tried to remove "noisy supervisions" that were generated by harsh augmentation on images. Han et al., [2] and Li et al., [3] also use distillation-like processes to learning noisy datasets.

[1] Label Refinery: Improving ImageNet Classification through Label Progression, Bagherinezhad  et al., 2018
[2] Co-teaching: Robust Training of Deep NeuralNetworks with Extremely Noisy Labels, Han et al., 2018
[3] Learning from Noisy Labels with Distillation, Li et al., 2017

**Experience Assessment:**

I have published one or two papers in this area.

**Review Assessment: Checking Correctness Of Derivations And Theory:**

I assessed the sensibility of the derivations and theory.

**Review Assessment: Checking Correctness Of Experiments:**

I assessed the sensibility of the experiments.

**Review Assessment: Thoroughness In Paper Reading:**

I read the paper at least twice and used my best judgement in assessing the paper.

---

> ### Author Response · Authors · 2019-11-07
> **Author Response**
>
> We thank the reviewer1 for his effort paid for reviewing our draft. It seems that most of the raised concerns are misunderstandings that can be resolved with the following clarification.
>
>  Regards Justification and explanation of the motivation
> > The explanation we give is “Distillation works due to the soft targets generated by the teacher network. Based on the observation that the overparameterized network can exactly fit the one-hot labels which contain no dark knowledge, we theoretically justify that early stopping is essential for an overparameterized teacher network to extract dark knowledge from the hard labels.” which appears in the introduction part paragraph 3,  contribution 1.1 paragraph 1 and conclusion part. We also have a toy example in Section 2.1 to support this point.
>
> Relationship between Section 2 and Section 3
> > Section 2 saying that without early stopping distillation can’t work, i.e. distillation is transferring the regularization effect of early stopping. A natural question is why we need distillation if it is just early stopping.  Thus in Section 3, we set up a distillation algorithm to show how distillation can enhance the regularization effect. Theoretically, we achieve a better bound and empirically we achieve a significant performance boost.
>
>
> Regards to the relationship between NTK and our theory.
>
> > Please go through section 3.5, all the subsection is discussing how the informative information can be enhanced and transferred, it is also the main idea used in the proof of the main theorem to achieve better bound than early stopping. Without NTK we can formalize the AIR of a neural network and gives out the theoretical guarantee. You can go through our proof, actually, the eigenspace of the NTK is the most important part.
>
>
> Regard claimed contributions
> > I have cited all the papers you listed and have discussed the relationship with it in section 3.1. Please go through it.  In short, the contribution we claim is
> We complete the teacher-student training in one generation but not introduce a further teacher/co-learner, which makes the training cheaper. [3] even using extra knowledge graph, focus on totally different point as our paper does.
> This paper aims to understand distillation **theoretically**. (to our knowledge) It’s the first one to consider seriously why distillation can help cleaning label noise and what’s the benefit using distillation, which is mentioned by reviewer 1 and reviewer 2, like “The paper is interesting and brings new theoretically thoughts to understand the distillation method. The relationship between early stopping and distillation can inspire the machine learning researchers to explore more about distillation both empirically and theoretically.” and “I find the self-distillation algorithm very interesting and it's nice that it can achieve zero l_2 loss w.r.t the correct labels.” (While the previous result was on convergence in 0-1 loss which is a much weaker result).
>
> Let me know whether these responses addressed your concern?

---

> > ### Comment · AnonReviewer3 · 2019-11-08
> > **Concerns are not resolved**
> >
> > Thank you for your response. However, I would like the authors to further elaborate on their responses. To clarify my concern, I am trying to find a good theoretical explanation on how "early stopping is essential for knowledge distillation" as claimed in the paper.
> >
> > Regards Justification and explanation of the motivation.
> > > I do not think the author's response is "justifications" for their main statement, which is what I have asked in the first place. From what I understand,  the author's statements that "early stopping is essential for knowledge distillation" is only supported in Figure 1. Note that the "introduction part paragraph 3,  contribution 1.1 paragraph 1 and conclusion part" pointed out by the author are not ***theoretical*** justifications.
> >
> > Relationship between Section 2 and Section 3
> > > I still do not see a good connection between Sections 2 and 3. The author claims that the purpose of Section 3 is to show how knowledge distillation can enhance the regularization effect of early stopping. This does not resolve the lack of justification behind the statement "early stopping is essential for knowledge distillation" in Section 2.
> >
> > Regards to the relationship between NTK and our theory
> > > If NTK is the essence for proof in Theorem 3, please state it in the main material of the paper for the comprehensibility of the paper. Furthermore, please (theoretically) demonstrate how the knowledge distillation process transfers the informative information, defined as "eigenspaces associated with the largest few eigenvalues of NTK" in the paper.
> >
> > Regard claimed contributions
> > > Thank you for adding the references as related work. I would also like to suggest comparing with [1],
> > where soft bootstrapping is quite similar to your algorithm.
> >
> >
> > --------- Additional Comment ---------
> > I also have the impression that the paper could improve its comprehensibility in general. I suggest fixing typos, e.g., "eigenspaces associated with the largest few eigenvalues of NTK" in page 4 and Table 1. with margin overflow. The title starting with DISTILLATION \approx EARLY STOPPING is also confusing since the main claim of the paper is "early stopping is essential for distillation", which does not mean two things are approximately same.
> >
> > [1] TRAINING DEEP NEURAL NETWORKSON NOISY LABELS WITH BOOTSTRAPPING, Reed et al., 2014

---

> > > ### Author Response · Authors · 2019-11-08
> > > **Justification of you comments**
> > >
> > > Thanks for your response. I think I can explain it in the following way
> > >
> > > Regards  "early stopping is essential for distillation"
> > > The **theoretical** justifications.
> > > > “Distillation works due to the soft targets generated by the teacher network. Based on the observation that the overparameterized network can exactly fit the one-hot labels which contain no dark knowledge, we theoretically justify that early stopping is essential for an overparameterized teacher network to extract dark knowledge from the hard labels.”
> > > > If you don't early stop the teacher network, proved by [1] and ton's of the following work, the teacher network will converge to the hard label provided by the dataset.  The objective function you define in the distillation is just the same as the original training process.  We can write it down as a serious theorem easily if you want.....
> > > (The theorem is just like if the teacher network is trained till converged, then the training of student network is the same as training without distillation.)
> > > (Consider that our loss is minmize_{NN} Loss(NN,label)+Loss(NN,Teacher), and now Teacher = label if you don't early stop the network which is proved by [1], then it's equal to  minmize_{NN} Loss(NN,label), which is the objective without distillation.)
> > > [`1] Du S S, Lee J D, Li H, et al. Gradient descent finds global minima of deep neural networks[J]. arXiv preprint arXiv:1811.03804, 2018.
> > >
> > > Regards section3.
> > > > Section 3 is not providing the evidence that "early stopping is essential for distillation". We are saying "distillation can work better than early stopping".
> > > >"early stopping is essential for distillation" is one of the main claims, seems that you are missing "distillation can work better than early stopping" is also an important claim. And this part is proved in section 3. (l2 convergence v.s. previous 0-1 loss convergence result.)
> > >
> > >
> > > Regards to the relationship between NTK and our theory
> > > > We have concluded NTK theory in section2...
> > > > The theoretical justification of "eigenspaces associated with the largest few eigenvalues of NTK"  informative information is from the data assumption. Without theoretical justification of ""eigenspaces associated with the largest few eigenvalues of NTK" is informative information" you can't prove the theoretical guarantee of recovering labels in section 3.
> > >
> > >
> > > > [1] is dealing with modeling the noise distribution as a matrix mapping model predictions to
> > > training labels. It's the case called "Pair flipping" but our paper is dealing with "symmetric noise".
> > > And why their method is similar to ours?
> > >
> > > Regards the title
> > > > We use DISTILLATION \approx EARLY STOPPING to express "distillation has the same kind of regularization effect introduced by early stopping" and "distillation can work better than early stopping"
> > >
> > >
> > > --------- Additional Comment ---------
> > > The reference is not **added**, all the reference you listed is already concluded in our first version of our paper before you reviewing the paper.
> > >
> > >
> > > Let me know if any of the concerns doesn't be justified.

---

> > > > ### Comment · AnonReviewer3 · 2019-11-08
> > > > **About theoretical justification**
> > > >
> > > > Thank you for your reply.
> > > >
> > > > I now agree that my comments on references are wrong. I am very sorry about missing this part.
> > > >
> > > > Based on the responses, I understand the author's justification behind the statement "early stopping is essential for knowledge distillation" as the following: if you do not early-stop the teacher, it would converge to a hard label and hence becomes training without using knowledge distillation.
> > > >
> > > > If this is the case, I do not think this is a significant enough result to state that "we theoretically justify that early stopping is essential for an overparameterized teacher network to extract dark knowledge from the hard labels." in Section 1.1 and I would suggest softening down this part.

---

> > > > > ### Author Response · Authors · 2019-11-08
> > > > > **About contribution**
> > > > >
> > > > > Thanks for your reply.
> > > > >
> > > > > The reason why we consider this is an important point is that this understanding enables us to transfer theory for early stopping to distillation and figure out the benefit of using distillation.  This understanding gives us the chance to form a theory for distillation (The first theory of distillation for deep learning)
> > > > >
> > > > > I'll rewrite section 1.1 as
> > > > > - early stopping is essential (I'll delete the word theoretically)
> > > > > - transfer theory of early stopping to distillation
> > > > > - distillation can work better than early stopping
> > > > > - based on our understanding, we designed a state-of-the-art clean label noise algorithm
> > > > >
> > > > > At the same time
> > > > > "distillation can work better than early stopping".
> > > > > is also a very important part in our paper.(l2 loss v.s. 0-1 loss convergence result and empirical result)
> > > > >
> > > > > The only paper we know to clean label noise theoretically using a neural network is  also an ICLR submission
> > > > > https://openreview.net/forum?id=Hke3gyHYwH&noteId=Hke3gyHYwH
> > > > > Comparing the results you can find out that our empirical result is much better. (the theory is very different, regression v.s. classification)
> > > > > p.s. Ridge regression is equivalent to kernel regression(also gradient descent in NTK regime) which is shown by many papers, one example is the one we cite
> > > > > [1] Yao Y, Rosasco L, Caponnetto A. On early stopping in gradient descent learning[J]. Constructive Approximation, 2007, 26(2): 289-315.
> > > > > Thus their result is equivalent to early stopping and that' why ours achieves better empirical result.
> > > > > We also cited their paper and discuss the in the first version of the draft.
> > > > >
> > > > > Let me know if you have any questions.

---

> > > > > ### Author Response · Authors · 2019-11-11
> > > > > **Follow up**
> > > > >
> > > > > It's just a follow up to confirm all of your concerns have been justified, in this case, will you change your score?

---

### Author Response · Authors · 2019-11-14
**Starting discussion**

We thank your effort to review our paper and contributive suggestions you raised. We also want your to take attention to our response.  Let us know what is the factor you are really concerning with.

---

### Decision · Program_Chairs · 2019-12-19

**Decision:**

Reject

**Comment:**

This paper tries to bridge early stopping and distillation.

1) In Section 2, the authors empirically show more distillation effect when early stopping.
2) In Section 3, the authors propose a new provable algorithm for training noisy labels.

In the discussion phase, all reviewers discussed a lot. In particular, a reviewer highlights the importance of Section 3. On the other hand, other reviewers pointed out "what is the role of Section 2", as the abstract/intro tends to emphasize the content of Section 2.

I mostly agree all pros and cons pointed out by reviewers. I agree that the paper proposed an interesting idea for refining noisy labels with theoretical guarantees. However, the major reason for my reject decision is that the current write-up is a bit below the borderline to be accepted considering the high standard of ICLR, e.g., many typos (what is the172norm in page 4?) and misleading intro/abstract/organization. In overall, it was also hard for me to read the paper. I do believe that the paper should be much improved if the authors make more significant editorial efforts considering a more broad range of readers.

I have additional suggestions for improving the paper, which I hope are useful.

* Put Section 3 earlier (i.e., put Section 2 later) and revise intro/abstract so that the reader can clearly understand what is the main contribution.
* Section 2.1 is weak to claim more distillation effect when early stopping. More experimental or theoretical study are necessary, e.g., you can control temperature parameter T of knowledge distillation to provide the "early stopping" effect without actual "early stopping" (the choice of T is not mentioned in the draft as it is the important hyper-parameter).
* More experimental supports for your algorithm should be desirable, e.g., consider more datasets, state-of-the-art baselines, noisy types, and neural architectures (e.g., NLP models).
* Softening some sentences for avoiding some potential over-claims to some readers.